# The network signature of constellation line figures

**Doina Bucur** *

Department of Computer Science, University of Twente, Enschede, The Netherlands

* d.bucur@utwente.nl

## Abstract

In traditional astronomies across the world, groups of stars in the night sky were linked into constellations—symbolic representations rich in meaning and with practical roles. In some sky cultures, constellations are represented as *line* (or connect-the-dot) *figures*, which are spatial networks drawn over the fixed background of stars. We analyse 1802 line figures from 56 sky cultures spanning all continents, in terms of their network, spatial, and brightness features, and ask what associations exist between these visual features and culture type or sky region. First, an embedded map of constellations is learnt, to show *clusters* of line figures. We then form the network of constellations (as linked by their similarity), to study how *similar* cultures are by computing their *assortativity* (or homophily) over the network. Finally, we measure the *diversity* (or entropy) index for the set of constellations drawn per sky region. Our results show distinct types of line figures, and that many folk astronomies with oral traditions have widespread similarities in constellation design, which do not align with cultural ancestry. In a minority of sky regions, certain line designs appear universal, but this is not the norm: in the majority of sky regions, the line geometries are diverse.

**Data Availability Statement:** Data public. Part of the data is part of the Stellarium code repository, https://github.com/Stellarium/stellarium-skycultures and the other part I made public already at https://github.com/doinab/constellation-lines (link also in the manuscript).

## 1 Introduction

For sky watchers through time, the night sky was a canvas to be filled with symbols. They designed *constellations* as groups of stars, which were named and assigned a practical utility or a background story. The constellation figures form a visual communication system in some ways similar to characters in a written script [1]; they are more or less complex in form, may or may not resemble the animal, human figure, or object that they were named after, and may or may not have been drawn similarly in unrelated cultures [2]. Constellations are now usually represented as *line figures*, with the stars connected by imaginary lines. There is great diversity among their shapes, sizes, and internal complexity across cultures—Fig 1 shows traditional Chinese [3, 4] and reconstructed ancient Babylonian constellations [4–7] for the same southern sky region: where the Chinese drew abstract, short and twisted chains, the Babylonians filled large surfaces with polygons and realistic human or animal figures.

We study 56 sky cultures across the world (located geographically in Fig 2), with line figures recorded in the literature. For some cultures (such as the Chinese and their area of influence), the line-figure style of representation dates to the beginning of their astronomical records [3].

**Funding:** The author(s) received no specific funding for this work.

**Competing interests:** The authors have declared that no competing interests exist.

For others (such as the Western sky culture and its ancestors), the line figures are the result of an evolutionary process from allegorical pictographs [8, 9] to now globally recognised figures [10, 11]. For yet other cultures (such as those native to the Americas), the line figures are often recent interpretations for unlined groups of stars [12–18].

We ask whether the geometry of constellations is unique to a sky culture (are humanoid-like figures characteristic only to Babylonia, and abstract figures only to China?), or to a type of culture (do cultures of Mesopotamian ancestry, or seafaring cultures, each have a characteristic geometric style?). We also ask whether certain line figures are universal in a sky region (were they driven by the star pattern, not by the sky watchers?). We investigate these questions in a unified way across cultures.

We first settle the technical terms, then draw the hypotheses.

**Sky culture** denotes the astronomical traditions of a society at a point in time. The cultures are treated equally (a Native American tribe is treated on par with Imperial China), regardless of the population size, the time of documentation (from before 0 AD to the 21st century), the author (the society as a collective, or a specific individual), or the number of line figures.

**Constellation or asterism** denotes a group of stars joined into a line figure, rather than the International Astronomical Union (IAU) definition [11], namely a bounded region of the sky. The difference between constellations and asterisms is the tendency for asterisms to be small in size.

**Line figure** denotes the dot-and-line representation of constellations. Other names include connect-the-dots or stick figures. Line figures are *spatial graphs*. The stars are *nodes*, at

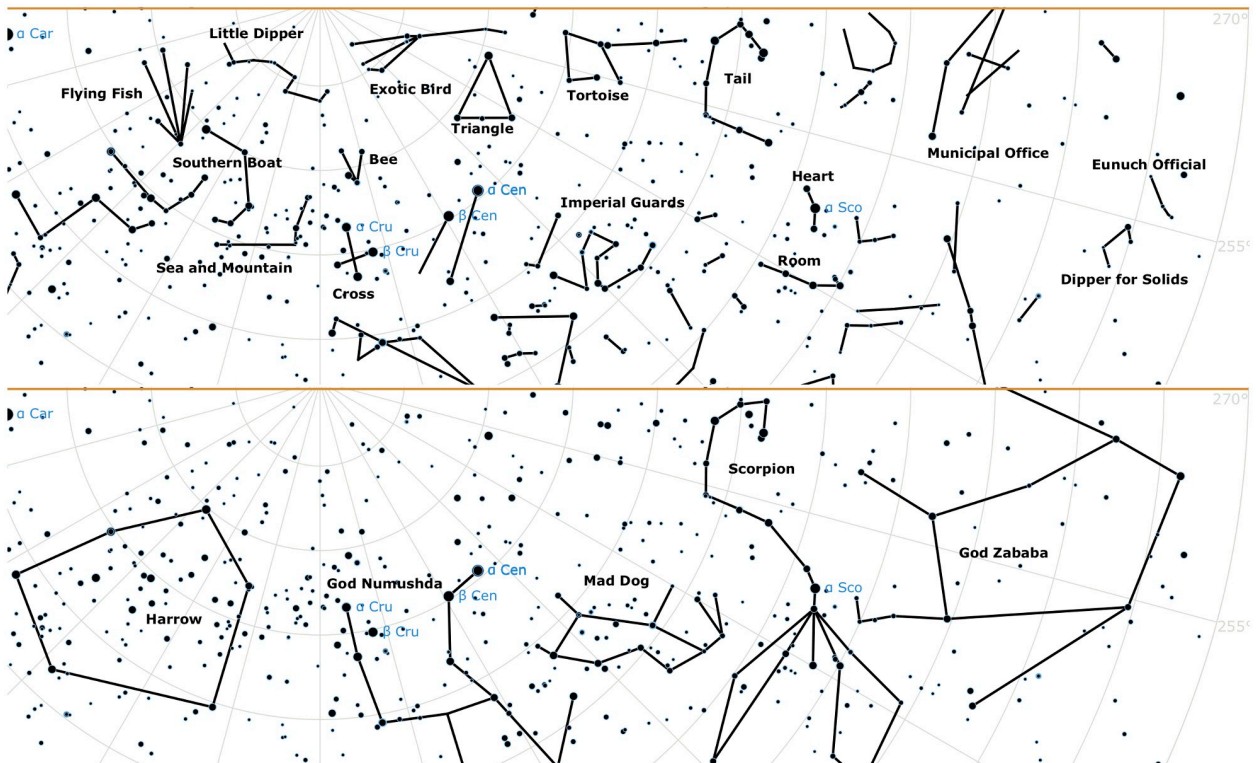

**Fig 1. The diversity of constellations line figures.** Traditional Chinese (top) and ancient Babylonian (bottom) constellations for the same southern sky: declinations [−90˚, 20˚], right ascensions [90˚, 270˚]. The choice of stars and lines differs. Some constellation names removed for clarity.

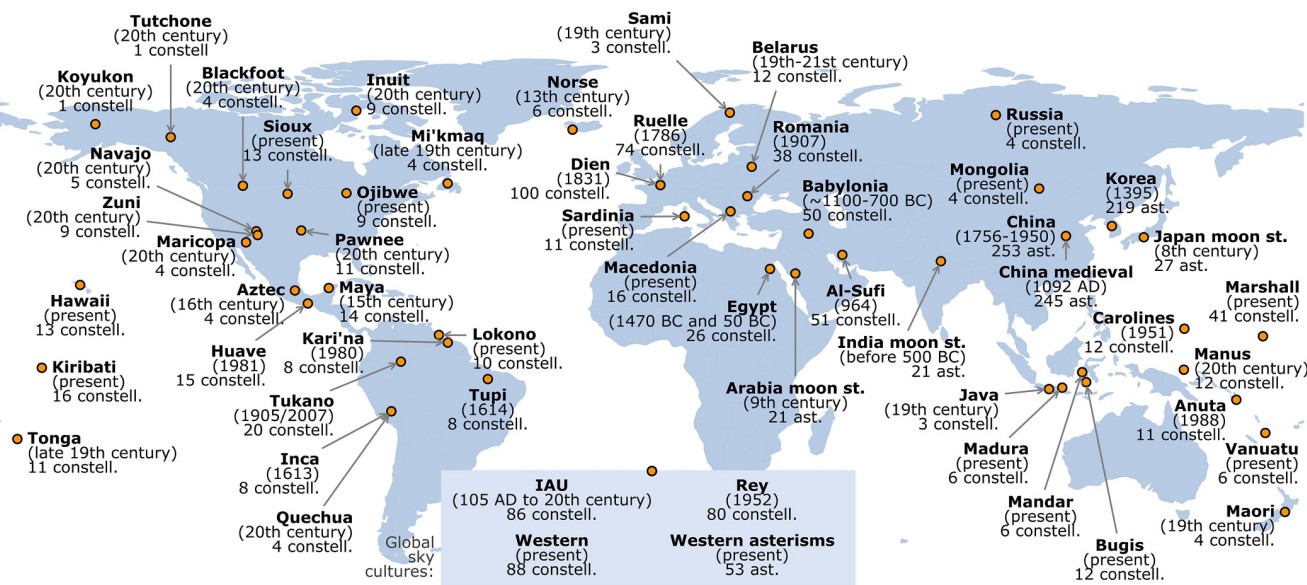

**Fig 2. The location of sky cultures.** The 56 cultures are shown with: name, the date of documentation, and the number of constellations (or asterisms) with at least one line. Sky cultures with global reach are highlighted at the bottom.

specific spherical coordinates on the sky, labelled with a magnitude. The *links* are arcs (or geodesics) between pairs of stars. The spatial graphs may or may not be spatially planar, or even connected (although the majority are).

**Visual signature** denotes here a set of 19 measurable, quantitative features (or statistics) of a line figure. These features include *network features* (the number of nodes, the size of components and cycles, statistics over node degrees, the connectivity of the spatial graph, and others), *spatial features* (the diameter of the figure, the length of its edges, the sharpness of the angles, and whether it is spatially planar), and *brightness* features (statistics over the star magnitudes).

We draw two hypotheses.

## [Hypothesis I] The type of sky culture associates with the visual signature of constellations

The type of a sky culture may relate with, and possible have determined, how constellations look in that culture, the way the complexity of characters in a written script may have been determined by the type of script [1]. We use mostly associative language ("associates with") rather than causal language ("determines" or "influences"), because a causal link cannot be proven. We study the following four culture types:

**I.1** The **culture** itself: Does each culture uniquely associate with the visual signature of its constellations? In other words, are its constellations not only homogeneous visually, but also distinct from those of other cultures, as the Chinese constellations are when compared to the Babylonian?

**I.2** Astronomical **literacy**: Do written (as opposed to oral) astronomical traditions associate with the visual signature of constellations from these traditions?

I.3 The **practical use** of constellations: Do constellations used as markers for open-sea navigation have a different visual signature than those used for political or religious astrology, or for time-keeping in agrarian and hunter-gatherer societies?

I.4 The **phylogeny** of the culture: Is a common ancestry of cultures associated with the visual signature of constellations from that ancestry?

The literature shows similarities between cultures based *only* on selected constellations: the Big Dipper asterism connects western Siberian with western N-American traditions, and Orion's Belt connects central Eurasian with (south)western N-American, perhaps Mesoamerican, and some Polynesian traditions [14, 19, 20]. No study quantified the visual complexity of line figures, nor the association between that and culture type, by which cultural (dis)similarities can be studied *at scale*.

Our **results** reveal a clear, global picture. Sky cultures across the world *cluster* in visual signature, with almost *no similarity* between three major clusters: (1) literate traditions of Chinese ancestry, (2) written traditions of Mesopotamian, Egyptian, and Greek ancestry, and (3) oral traditions across the globe. This lack of connection can be explained by the largely independent development of these traditions. More surprisingly, weak but widespread *similarities* connect most *oral folk traditions across five geographical regions* (N- and S-America, Europe, Asia, and the Pacific) into a dense cluster. This finding cannot be explained by a common ancestry, so it implies that cross-cultural principles of forming constellations may exist (and remain to be studied).

## [Hypothesis II] The region of the sky associates with the visual signature of constellations across cultures

Having multi-cultural constellation data over the same sky allows us to make a distinction between the background data (e.g., the star pattern around the celestial north pole) and the foreground data (i.e., the line figures drawn in that region by various cultures). Since the background data is fixed, a complementary hypothesis arises: the sky background itself may be the significant driver of constellation shapes, overpowering the role of the culture typology. Variants of the Big Dipper, Orion, and other star groups recur across cultures [19, 21], but does diversity remain in the design of line figures? We answer the question:

II The **sky region**: Is there recurrent universality in the visual signature of constellations per region of the sky, or, on the contrary, diversity?

Previous studies stated some cognitive basis for forming star groups without lines (for 30 constellations [22], and for 27 cultures [21]), finding that perceptual grouping can explain part of the popular asterisms.

Our **results** point to a much more complex picture over more data (56 sky cultures). We find that, among the visual signatures of constellations from popular sky regions, *diversity is more likely than universality*: the majority of sky regions have *high diversity*, with particularly high scores for the regions overlapping IAU Pegasus, Andromeda, Canis Major, Orion, Sagittarius, and Cygnus. On the other hand, low diversity, or nearly *universal constellation design* across cultures, is found in a minority of sky regions overlapping IAU Cassiopeia, Corona Borealis, Leo, Ursa Major, and particularly Scorpius. This implies that some patterns of stars are indeed conducive to universal design, but also that these patterns (such as the long chain of bright stars in IAU Scorpius) occur quite rarely in reality.

We provide our results in Sec. 2, a discussion and conclusion in Sec. 3, details on the data in Sec. 4 and on the method in Sec. 5.

## 2 Results

### 2.1 Statistics of constellation features

The constellations are spatial networks. Their features vary greatly, both within and across cultures. Fig 3 shows the culture size, and average and standard deviation of four **constellation features** per sky culture.

The features in Fig 3 are selected from a set of 19 such features, denoted $s_1$ to $s_{19}$ and listed below in brief (for details on their definition and meaning, see Sec. 5.1). *Network constellation features* are structural statistics of the line figure:

$s_1$ the **number of links**;

$s_2$, $s_3$ the **maximum** and **average degree**;

$s_4$ the **clustering coefficient**;

$s_5$ the **maximum core number**;

$s_6$, $s_7$ the **number of basic cycles**, and the size of the **largest basic cycle**;

$s_8$ the **number of connected components** (CCs);

$s_9$, $s_{10}$ the average **link diameter** and shortest path among the CCs;

$s_{11}$ the **link connectivity**.

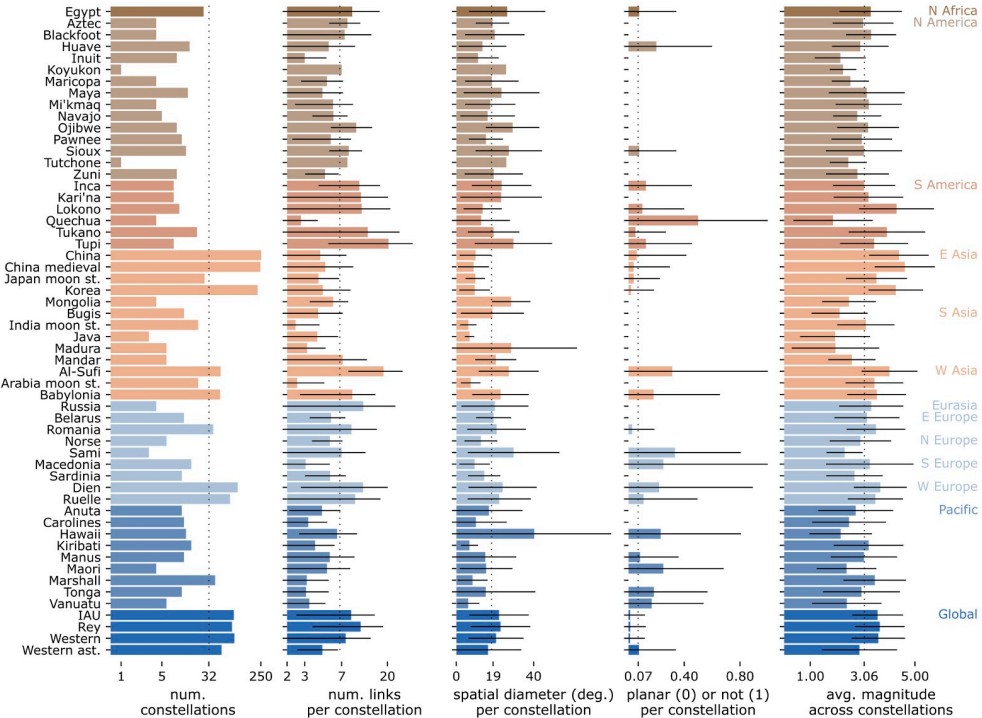

**Fig 3. Constellations features aggregated per sky culture.** The culture size (constellation count) is on the left. Four constellation features ($s_1$, $s_{12}$, $s_{16}$, and $s_{17}$) are then shown via their averages and standard deviations per culture. The global average of each statistic is marked with a dotted line. The horizontal scales for the first two statistics are logarithmic, and the rest linear.

*Spatial constellation features* capture geometric statistics:

$s_{12}$, $s_{13}$ the **spatial diameter** of the constellation, and the **average link length** (in degrees on the celestial sphere, from the point of view of an observer);

$s_{14}$, $s_{15}$ the **sharpest** and the **average angle** formed by any two links incident at any star (both in degrees);

$s_{16}$ whether the spatial network is **planar** or not.

*Brightness constellation features* measure basic statistics on star magnitudes:

$s_{17}$, $s_{18}$, $s_{19}$ the **average, minimum**, and **maximum star magnitude**.

The standard deviations in Fig 3 tend to be large, so the cultures are internally diverse. We thus expect that, if an association exists between the individual sky culture and the visual signature of constellations, it is weak. On the other hand, there are also clear distinctions between sky cultures. Five out of six S-American cultures have very large constellations (an average of 20.50 links per constellation for Tupi, and 12.40 for Tukano), compared to an average of 6.72 links per constellation across all cultures. All moon-station asterisms have few links (averages between 2.42 and 4.11 links per culture). Large E-Asian sky cultures (China and Korea) are also below average: averages between 4.28 and 4.82 links per culture. In terms of spatial diameter, it is not only sky cultures with many links per constellation that also have large constellation diameters: the largest average spatial diameter is Hawaii (40.27˚, compared to a global average of 18.25˚), although the same culture only has a below average number of links per constellation. The fraction of non-planar constellations is close to zero in N America and S Asia, but occasionally high for other cultures. In N America, S Asia, and the Pacific, cultures consistently use brighter (lower-magnitude) stars. This raises the expectation that an association may exist between cultures (when aggregated by type, such as by common ancestry, correlated to geographic location) and visual signature.

## 2.2 A clustered map of constellations

We first build an intuition about the types of line figures present. For this, we project all 1802 constellations across sky cultures into a two-dimensional "map of constellations", which is learnt by t-SNE embedding from the 19 original constellation features. The embedding has a very high trustworthiness score of 0.98 out of 1, meaning that the neighbours of each constellation by Euclidean distance are largely preserved between the original 19 and the final two dimensions. (The method and its evaluation are detailed in Sec. 5.2.) Fig 4 shows and interprets this map.

The map has rich internal structure. The orientation of the embedding, the measurement units, and the concrete values on the axes are not meaningful [23, 24] so are not shown; instead, it is the gradients of each of the 19 original constellation features which help to interpret these reduced dimensions. Twelve of them are overlaid on the embedding, at the top of Fig 4. There are distinct "islands" (or clusters) of constellations, and also local and global gradients for the constellation features.

**The embedding of constellation features**. Constellations with a large number of links ($s_1$) are located right of centre and are distributed among a number of clusters; those with only one link (the simplest possible shape) form an isolated cluster on the left. The peaks for the average degree ($s_3$), the number of cycles ($s_6$), and the link diameter ($s_9$) are close to the top-right corner, and that for clustering ($s_4$) is at the extreme right, with smooth gradients for all these features across the embedding. The maximum core number (not shown) is low and only takes

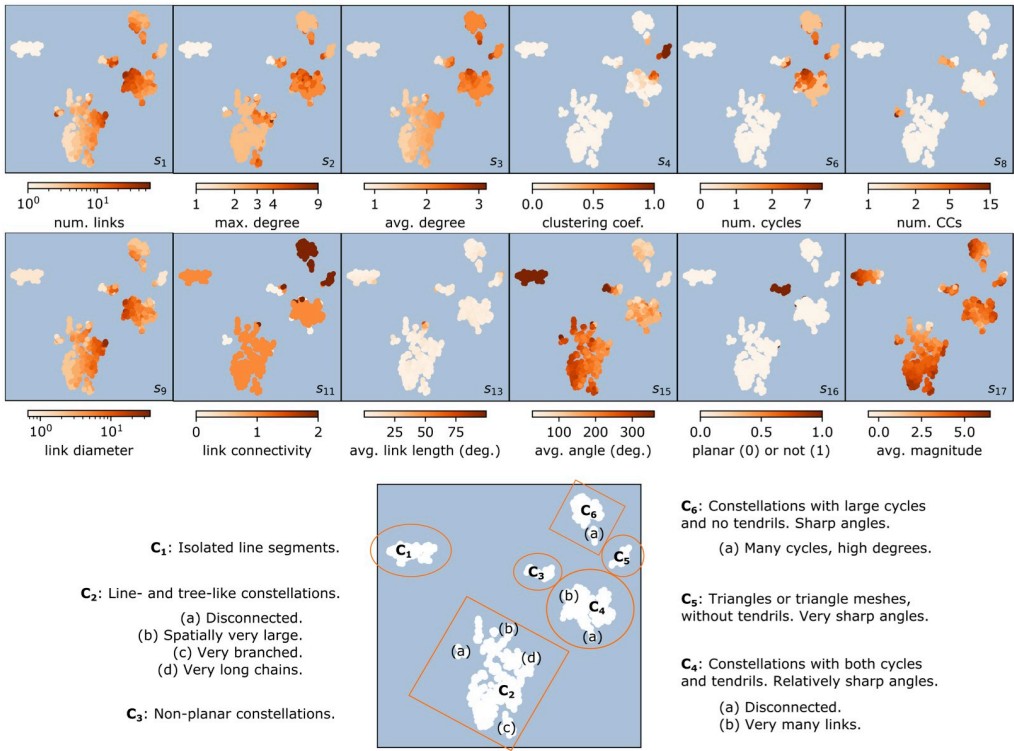

**Fig 4. The map of constellation features in two embedded dimensions.** All plots show the same embedding. One point represents one constellation. **(top)** The gradient of each constellation feature, as projected in the low-dimension space. Twelve constellation features are shown. **(bottom)** A summary of the clusters.

values 1 and 2; almost all constellations with a 2-core are embedded in the top-right quadrant. There are relatively few disconnected constellations ($s_8$), and they are embedded close to connected constellations which have otherwise similar statistics in other dimensions. The average angle formed by two links in a constellation ($s_{15}$) also has a global gradient, from 360° (meaning no angles) in the leftmost cluster, to sharp angles throughout the rightmost. Almost all non-planar constellations ($s_{16}$) are isolated from the rest, in a central cluster. Many statistics have local gradients; in particular, the average magnitude ($s_{17}$) has a distinct gradient within many of the clusters.

**Six clusters of constellations**. Fig 4 (bottom) summarises intuitively the clusters of line figures. (They correspond to the strongly connected components of the nearest-neighbour network of constellations described in Sec. 5.3).

Cluster $C_1$ (the smallest figures: single links) comprises 12% of all constellations. $C_2$ (line- and tree-like constellations) is the largest cluster at 46% and has small subclusters, e.g., disconnected line figures in (a). $C_3$ (4% of the total) groups almost all of the non-planar constellations; besides their non-planarity, these constellations are otherwise diverse in shape. $C_4$ (large constellations with complex internal structures of both cycles and tendrils) is the second-largest cluster with 21%. $C_5$ (triangles and triangular meshes, no tendrils) groups 4% of the figures. Finally, $C_6$ (cycles larger than three lines, no tendrils) comprises 12%.

In the embedding, constellations in different clusters, but with some common features, remain close. For example, constellations in $C_3$ with few links ($s_1$) are oriented towards $C_1$, and those with many links towards $C_4$. The clusters have internal gradients in feature values. We show where well known IAU constellations, and less known constellations from other

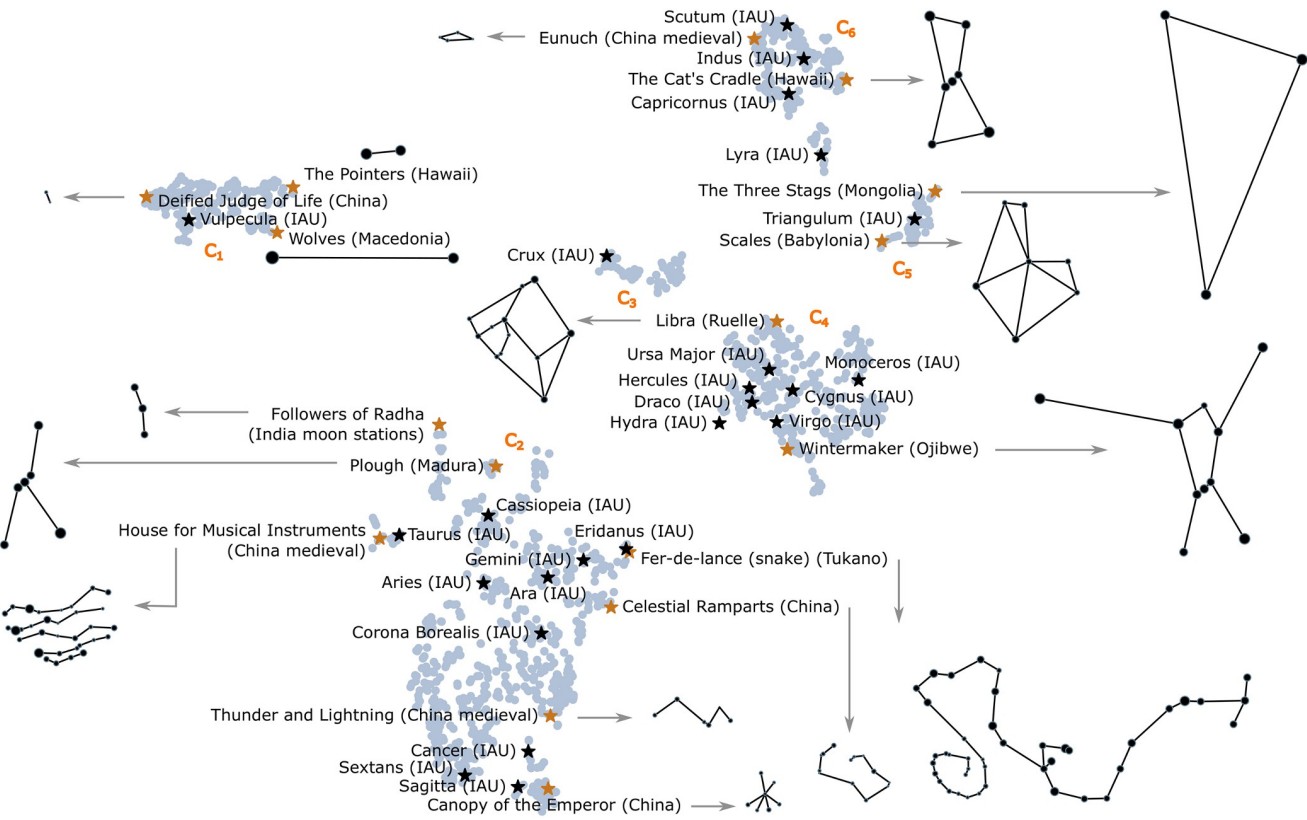

**Fig 5. Examples of constellations over the embedding.** In the background, in light blue, the embedding of all constellations, the same as in Fig 4. In the foreground, we show example constellations: (1) black markers for IAU constellations [11]; (2) orange markers for other cultures.

cultures, are embedded, in Fig 5: black stars mark some of the IAU constellations [11], while orange ones mark some constellations from other cultures. Their line figures are shown to scale.

## 2.3 [Hypothesis I] The type of sky culture associates with the visual signature of constellations

The questions we posed can now be answered by introducing the types of cultures, and measuring the extent to which they have unique, characteristic visual signatures. We do these measurements over the original set of 19 constellation features, and use the two-dimensional embedding from Sec. 2.2 to explain the findings intuitively.

For this hypothesis, we require a score which reaches its maximum value 1 for a strong association between a culture type and visual signature. For example, for question **I.1**, value 1 would signal that all constellations from a culture have a unique geometry different from that of all other cultures, or that the visual signatures *segregate* along cultural lines. The score must instead be 0 when the culture types mix randomly, or are de-segregated. A suitable score is the **assortativity coefficient** $r$ by a discrete node attribute [25]. This is computed over the *directed, nearest-neighbour network of constellations*: from each constellation, outlinks exist to its nearest neighbours by Euclidean distance. $r$ is accompanied by its *expected statistical error*, denoted $\sigma_r$. We also require a **similarity metric** $\Delta$ between any two culture types, which is positive if merging the two raises assortativity, or makes the mixing *less* random, so signals similarity. On the contrary, the similarity metric is negative if merging the two makes the mixing *more*

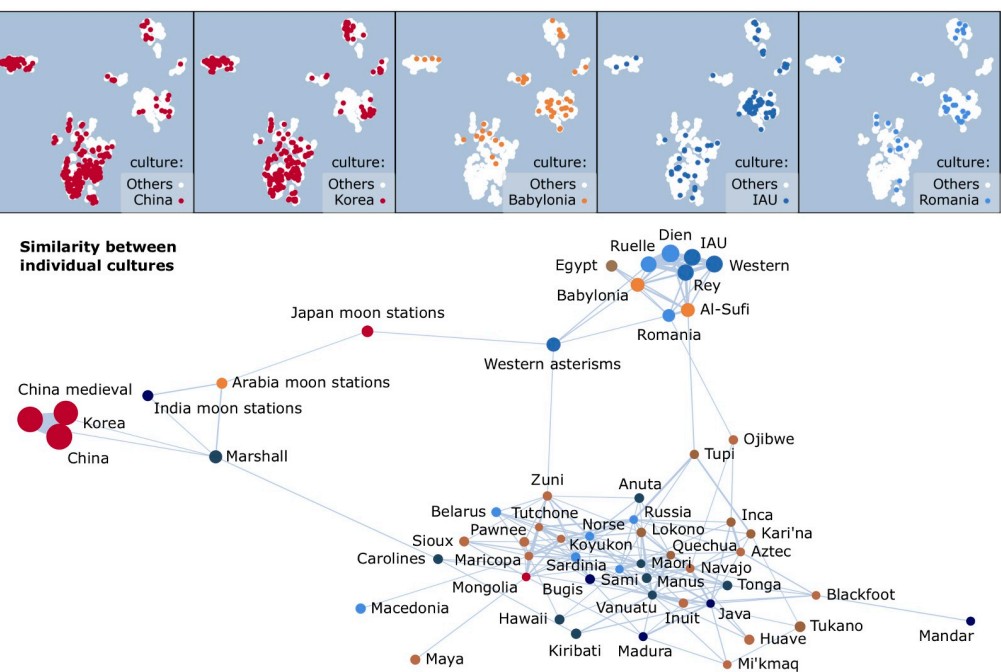

**Fig 6. Visual signatures by culture (question I.1). (top)** Constellations from example cultures are shown in the foreground, over the background of all other constellations. **(bottom)** The similarity graph for cultures. The node size is proportional to the number of constellations per culture, and the edge width to the similarity Δ.

random. This similarity metric is a statistic defined in this work. (Sec. 5.3 details the assortativity and similarity metrics.) We build a graph of similarities between culture types at each question, drawing links weighted by Δ.

**Question I.1. The culture itself.** *Cultures only weakly associate with a unique visual signature.* If the culture itself were to have a characteristic geometry, then constellations from the same culture would be (1) internally homogeneous, or close neighbours in signature, but also (2) externally heterogeneous, or apart from constellations of other cultures. When the predictor is the culture itself, this is only weakly the case (the assortativity is positive but low, $r = 0.079$ with $\sigma_r = 10^{-3}$). Although in some cultures many constellations are visually similar (and would be embedded closely together), in those cases there are similarities also with other cultures. This is shown for five large sky cultures in Fig 6 (top), where each plot emphasises the embedding of a single culture. The E-Asian cultures (China and Korea) are relatively homogeneous (with many constellations in clusters $C_1$ and $C_2$), but are also similar cross-culture. The same holds to a lesser extent for the three Babylonian and Western cultures, whose focal cluster is $C_4$. These similarities are expected, and may be naturally explained by a common ancestry. The constellations of many other sky cultures (not shown in the figure) are scattered in the embedding, so not even internally homogeneous.

We also zoom in locally, on one culture at a time, to assess how that culture mixes with all other cultures taken as one. The Chinese cultures and Charles Dien's 1831 star chart are the most distinct in geometry ($r = 0.141$ and $r = 0.140$, respectively). Other cultures are more weakly distinct ($r > 0.050$): Al-Sufi, Korea, Ruelle, Rey, IAU, and Western asterisms—all of the above with $\sigma_r < 10^{-3}$. The other cultures mix almost randomly with all others. We thus conclude that, with few exceptions, the 56 individual cultures have cross-culture commonalities in constellation design. In the following, we locate these commonalities.

*Culture similarity clusters do not always follow phylogeny.* The graph of similarities Δ computed between all pairs of cultures shows clear 'families' of cultures. Fig 6 (bottom) draws this graph. Only positive values for Δ are used (namely, similarities rather than dissimilarities), and the graph is laid out with a force-directed layout guided by the similarity as edge weight. The strongest pairwise similarity is unsurprisingly between China and medieval China (Δ = 0.097). The lowest similarity drawn in the figure is Δ = 0.005, to sparsify the graph.

China and Korea form their own cluster, on the left of Fig 6 (bottom). Another distinct cluster (at the top) is formed by the 'classical' cultures close to the ancestral roots of Western constellations (IAU and its derivations, plus Al-Sufi, Babylonia, and Egypt). Romania is the only folk culture similar to these classical cultures: it contains the same style of large, intricate constellations. These clusters follow ancestry, so are not surprising: the cultures of Chinese origin are strongly related, as expected (with the exception of the Japanese moon stations), and so are some of those with Greek and Mesopotamian origin. The asterisms of the three moon-station cultures are similar, but isolated from all others except for some similarity with the Western asterisms and the constellations of the Marshall Islands.

Surprisingly, the largest cluster (at the bottom) is formed by mostly folk astronomies with oral traditions, covering all continents. Many European folk cultures (also of Greek influence) are *as similar* with N-American, S-American, Austronesian, and Polynesian cultures as they are among themselves. We thus expect that some phylogenies will measure to be similar despite the geographical distance and lack of known cross-influence. Later, question **I.4** will test phylogeny as a predictor.

**Question I.2. Astronomical literacy.** A minority of the sky cultures in this study have written astronomies (marked in column **type** in Table 1 in Sec. 4 on the data). In these cases, the constellations were documented early on a longer-lasting medium, such as a codex, chart, book, stone or clay tablets. Unsurprisingly, the number of surviving constellations from written cultures is higher: the 30% of written cultures hold 78% of the constellations.

*Oral astronomies use brighter stars.* The type of astronomical literacy positively associates with the visual signature of constellations ($r = 0.234$ with $\sigma_r = 2 \cdot 10^{-3}$). This means that constellations issued from oral cultures can be statistically distinguished, to an extent, from those of written cultures. We test this by also training a Support-Vector classifier with a nonlinear kernel and balancing class weights, for the two classes. The classifier confirms this (balanced accuracy 0.75).

The positive association is explained by the characteristics of oral astronomies, most of which formed the largest similarity cluster in Fig 6 (bottom). The oral group consists of some Eurasian, almost all native American, Austronesian, and Polynesian cultures. Their constellations are present in all clusters of the embedding, but preferentially in regions where *brighter stars* are used—towards the centre of the embedding, as shown by the average magnitude ($s_{17}$) in Fig 4. The written cultures, on the other hand, do not have this preference: they include cultures of both Chinese and Western origins, which are complementary in visual signatures, and use all star magnitudes. The classifier confirms that brightness is crucial to distinguish oral from written constellations: the most important feature is the *average star magnitude*, $s_{17}$. The F1-score (the harmonic mean of the precision and recall) is higher for the written (0.86) than for the oral constellations (0.60), since faint constellations can be correctly assigned to written cultures, but bright constellations may belong to either class.

A caveat to this result is that this association may be due to the limitations of data collection from oral cultures (specifically, a possible bias towards recalling brighter constellations, as per Sec. 4).

**I.3. The practical use of constellations.** Only four cultures have *political divination* as practical use (column **type** in Table 1), but these comprise 41% of the constellations. They

**Table 1. Summary of sky cultures.**

| location | id. | an. | sky culture | timestamp | source | | type | count | references |
|---|---|---|---|---|---|---|---|---|---|
| Global | I | G | **IAU** | 105 AD–20th c. | standard | w | re,nv | 86 | [4, 11] |
| Global | | I | **Rey** | 1952 | book | w | re,nv | 80 | [4, 10] |
| Global | | I | **Western** | present | dataset | w | re,nv | 88 | [4] |
| Global | | I | **Western asterisms** | present | dataset | w | re,nv | 53 | [4] |
| N Africa | | | **Egypt** | 1470, 50 BC | carving,paper | w | re | 26 | [4, 28] |
| W Asia | M | | **Babylonia** | 1100-700 BC | tablet,papers | w | re | 50 | [4–7] |
| W Asia | | In | **Arabia moon st**. | 9th c. | book,paper | w | re | 21 | [4, 29] |
| W Asia | | G | **Al-Sufi** | 964 AD | book,dataset | w | re,nv | 51 | [4, 30] |
| S Asia | In | | **India moon st**. | < 500 BC | book,dataset | w | re | 21 | [4, 31] |
| S Asia | | A | **Bugis** | present | papers | o | nv | 12 | [32, 33] |
| S Asia | | A | **Java** | 19th c. | book,paper | o | fo | 3 | [34, 35] |
| S Asia | | A | **Madura** | present | paper | o | nv | 6 | [36] |
| S Asia | | A | **Mandar** | present | paper | o | nv | 6 | [32] |
| E Asia | C | | **China medieval** | 1092 AD | chart,book | w | po | 245 | [3, 4] |
| E Asia | | C | **China** | 1756-1950 | chart,book | w | po | 253 | [3, 4] |
| E Asia | | C | **Korea** | 1395 | chart,dataset | w | po | 219 | [4] |
| E Asia | | C | **Japan moon st**. | 8th c. | chart,paper | w | po | 27 | [4, 37] |
| E Asia | | I | **Mongolia** | present | dataset | o | fo | 4 | [4] |
| Eurasia | | I | **Russia** | present | book | o | fo | 4 | [38] |
| W Europe | | I | **Ruelle** | 1786 | chart | w | re,nv | 74 | [39] |
| W Europe | | I | **Dien** | 1831 | chart | w | re,nv | 100 | [40] |
| E Europe | | I | **Belarus** | 19th-21st c. | paper | o | fo | 12 | [4, 41] |
| E Europe | | I | **Romania** | 1907 | book,exhibition | o | fo | 38 | [4, 42, 43] |
| S Europe | | I | **Macedonia** | present | paper | o | fo | 16 | [4, 44] |
| S Europe | | I | **Sardinia** | present | dataset | o | fo | 11 | [4] |
| N Europe | | I | **Norse** | 13th c. | verse,book,dataset | o | nv | 6 | [4, 45] |
| N Europe | | | **Sami** | 19th c. | book | o | fo | 3 | [4, 46] |
| N America | | nA | **Maya** | 15th c. | codex,books | w | re | 14 | [4, 47, 48] |
| N America | | nA | **Aztec** | 16th c. | codices,book | w | re | 4 | [4, 49–51] |
| N America | | nA | **Huave** | 1981 | paper | o | fo | 15 | [52] |
| N America | | nA | **Inuit** | 20th c. | book,dataset | o | fo | 9 | [4, 53] |
| N America | | nA | **Koyukon** | 20th c. | book | o | fo | 1 | [14] |
| N America | | nA | **Tutchone** | 20th c. | book | o | fo | 1 | [14] |
| N America | | nA | **Mi'kmaq** | late 19th c. | book | o | fo | 4 | [14] |
| N America | | nA | **Ojibwe** | present | book | o | fo | 9 | [4, 16] |
| N America | | nA | **Blackfoot** | 20th c. | book | o | fo | 4 | [14] |
| N America | | nA | **Pawnee** | 20th c. | book | o | fo | 11 | [14] |
| N America | | nA | **Sioux** | present | books | o | fo | 13 | [4, 14, 15] |
| N America | | nA | **Maricopa** | 20th c. | book | o | fo | 4 | [14] |
| N America | | nA | **Navajo** | 20th c. | book | o | fo | 5 | [14] |
| N America | | nA | **Zuni** | 20th c. | book | o | fo | 9 | [14] |
| S America | | sA | **Inca** | 1613 | book | o | fo | 8 | [54] |
| S America | | sA | **Kari'na** | 1980 | paper | o | fo | 8 | [13] |
| S America | | sA | **Lokono** | present | dataset | o | fo | 10 | [4, 18] |
| S America | | sA | **Quechua** | 20th c. | paper | o | fo | 4 | [55] |
| S America | | sA | **Tukano** | 1905/2007 | book/thesis | o | fo | 20 | [4, 17, 56] |
| S America | | sA | **Tupi** | 1614 | book,papers | o | fo | 8 | [4, 12, 57, 58] |

**Table 1.** (Continued)

| location | id. | an. | sky culture | timestamp | source | | type | count | references |
|---|---|---|---|---|---|---|---|---|---|
| Pacific | | P | **Anuta** | 1998 | book | o | nv | 11 | [4, 59] |
| Pacific | | P | **Carolines** | 1951 | paper | o | nv | 12 | [60, 61] |
| Pacific | | P | **Hawaii** | present | website,dataset | o | nv | 13 | [4, 62] |
| Pacific | | P | **Kiribati** | present | dictionary | o | nv | 16 | [63] |
| Pacific | | P | **Manus** | 20th c. | paper | o | nv | 12 | [64] |
| Pacific | | P | **Maori** | 19th c. | paper | o | nv | 4 | [4, 65] |
| Pacific | | P | **Marshall** | present | dictionary | o | nv | 41 | [66] |
| Pacific | | P | **Tonga** | late 19th c. | paper | o | nv | 11 | [4, 67] |
| Pacific | | P | **Vanuatu** | present | website,dataset | o | fo | 6 | [4, 68] |

**id**. provides an identifier, and **an**. ancestry. Under **type**, knowledge transmission is marked as written (w) or oral (o). The (main) practical use(s) (at the time of origin) are for: navigation (nv), religious (re) or political (po) divination, or folk/agrarian/hunter-gatherer time-keeping on land (fo). The **count** includes constellations with at least one line.

include the distinct Chinese-Korean similarity cluster (from question **I.1**, shown in Fig 6). We thus expect that political divination retains a unique visual signature.

There are three other constellation uses in the dataset (16% *folk/agrarian/hunter-gatherer time-keeping or orientation on land*, 17% *navigation*, and 18% *religious divination*, with the remaining uncategorised). Whether or not they also have an association with the visual signature is not clear from the answer to question **I.1** alone. In particular, the Western cultures are mixed: they contain some constellations with an originally religious role (those inherited from Mesopotamia via the Greek [8, 26]), and some originally designed by navigators around the northern pole, on the celestial equator, and in the southern skies [9, 26, 27]. The practical use of the same star group also changed with the evolution of cultures: while Ursa Major was a navigator's constellation in the Greek tradition [9], its Big Dipper asterism is now part of many agrarian and hunter-gatherer folk cultures [19]. Only the seafaring cultures (of Austronesian, Polynesian ancestry) are geographically apart from most others, so some uniqueness in visual signature is expected.

*Navigation, religious, and folk constellations are similar*. For question **I.3**, the result only partly follows expectations. The practical use positively associates with the visual signature of constellations ($r = 0.238$), but this is mainly due a characteristic political-use signature ($r = 0.439$ when testing the mixing of political use with all other uses). ($\sigma_r < 3 \cdot 10^{-3}$ in both cases.) The location of political-use constellations over the embedding (Fig 7, left) shows the expected segregation of this use away from the centre of the embedding. On the other hand, all other uses (also in Fig 7) appear similar, so mix almost randomly. The similarity graph is in the

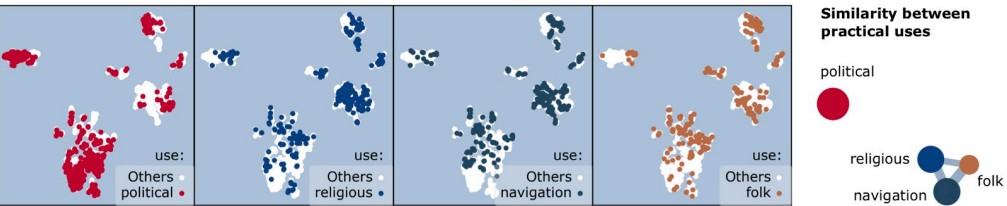

**Fig 7. Visual signatures by practical use (question I.3). (top)** Constellations per use are shown in the foreground, over the background of all other constellations. **(bottom)** The similarity graph between practical uses. The node size is proportional to the number of constellations per use, and the edge width to the similarity Δ.

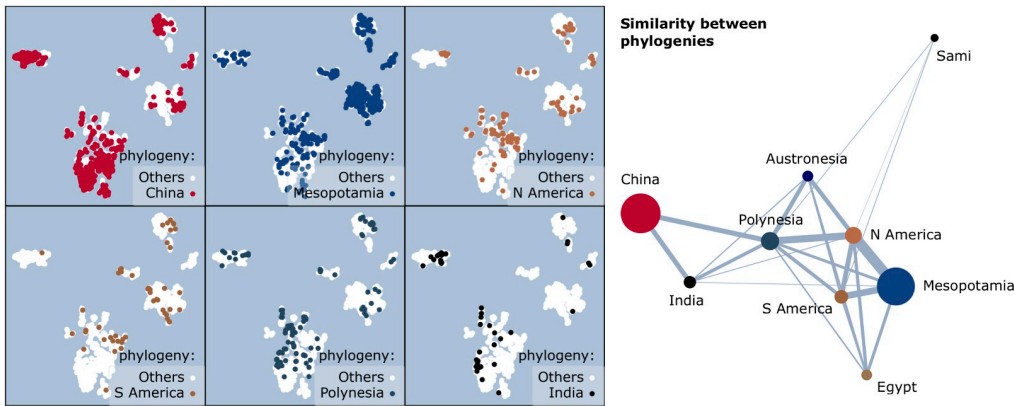

**Fig 8. Visual signatures by phylogeny (question I.4). (left)** Constellations with common phylogeny are shown in the foreground, over the background of all other constellations. The smallest three phylogenies are not shown: Sami (3 constellations), Egypt (26), and Austronesia (27). **(right)** The similarity graph between phylogenies. The node size is proportional to the number of constellations per phylogeny, and the edge width to the similarity Δ.

same figure, on the right. Only three positive similarity links exist, and they are comparable in value: the strongest is between navigation and folk constellations (Δ = 0.069).

We thus conclude that (1) as expected, political constellations have a strong associated signature, but (2) unexpectedly, all other use culture types including seafaring yield similar line figures.

**I.4. The phylogeny of the culture.** The 56 sky cultures are grouped into ancestry groups (by column **an**. in Table 1). The Mesopotamian ancestry includes Babylonia, and all descendants of Greek and IAU cultures. The Sami and Egyptian cultures have no known ancestry, and each forms their own group. Of the resulting nine groups, the largest six are shown in Fig 8.

For phylogeny as a predictor, we obtain a positive association with visual signature ($r = 0.291$ with $\sigma_r = 2 \cdot 10^{-3}$). However, this is due to a strong association for only some of the phylogenies: Chinese and Mesopotamian.

*Chinese and Mesopotamian ancestries have characteristic signatures*. Constellations of Chinese ancestry self-group to a large extent (Fig 8, left), and dominate the constellation clusters with the *simplest visual shapes over faint stars*: a large part of $C_1$ (isolated line segments), a large part of $C_2$ (line- and tree-like constellations), and part of $C_6$ (single-cycle constellations) —all with relatively faint stars. Chinese ancestry mixes relatively little with all others ($r = 0.440$), as expected from the answer to question **I.3**. Constellations of Mesopotamian ancestry self-group to a lesser extent ($r = 0.331$), and dominate the clusters defined, on the contrary, by the *most complex visual shapes over bright stars*: $C_4$ (constellations with both cycles and tendrils), $C_5$ (triangular meshes), and part of $C_6$ (constellations with cycles larger than a triangle)—all with relatively bright stars. The N-American ancestry overlaps with the Mesopotamian (draws a diversity of shapes over bright stars), so has less of a distinct signature ($r = 0.057$). The Polynesian has some commonalities with all other ancestries ($r = 0.077$). ($\sigma_r < 3 \cdot 10^{-3}$ in all cases.)

*Non-Chinese ancestries are similar, and Polynesia is a bridge*. The similarity graph between phylogenies (Fig 8, right) draws all existing edges with positive similarity values. The strongest pairwise similarity is between Mesopotamian and N-American ancestries (Δ = 0.037). In the similarity graph, regions of similarity are apparent: a possible Chinese zone of influence (with similarities with India and Polynesia), but also a tight cluster of six phylogenies from all

continents, with Polynesia forming a bridge between the two. If the Chinese zone of influence may be explained by geographical vicinity, the cluster of six phylogenies cannot. The outlier Sami culture (with only three constellations) bares some resemblance with Polynesian and American cultures. However, it is not internally homogeneous: the three constellations have little similarity among themselves, which leads to weak assortativity for this culture in any test.

This similarity graph signals that complex visual shapes over bright stars, using cycles, tendrils, and combinations, may be a natural, universal preference in sky cultures outside the zone of Chinese influence.

## 2.4 [Hypothesis II] The region of the sky associates with the visual signature of constellations across cultures

Some stars are much more frequently linked into constellations than others. From among the stars present in this dataset, 72 stars are present in 22 or more constellations each. Occasionally, a culture has constellation variants, or simply different constellations which overlap; in these cases, the same star is present in two or more constellations of the same culture. This is relatively rare; the majority of constellations per root star come from different cultures. We take these stars as *root stars*, each defining a sky region. When two stars are close in the sky, their constellation sets overlap, but are rarely identical. The most frequent star is ζ Ori, in Orion (IAU), with 62 constellation variants; outside the belt of Orion, the next most frequent star is β UMa, with 44. We measure the diversity for each of the 72 root stars.

Over the embedding, we have previously delimited clusters of visual signatures (Fig 4). Although they are of unequal sizes, they have significant presence from the root stars: even in the smallest clusters, $C_3$ and $C_5$, the root stars have 48 and 16 constellations, respectively. Almost all root stars are present in the largest clusters, $C_2$ and $C_4$. We compute the Shannon entropy or **diversity index** $H$ per sky region defined by root star, over these six clusters (Sec. 5.4 details this method).

We provide the results in Fig 9. Per root star, this shows the diversity index $H$ among all constellations of that star (on the left axis), and the number of constellations with that star (on the right axis). The dotted line marks the middle of the diversity range, 0.5, and the diversity markers are coloured differently above and below this; although this middle value has no particular meaning, it helps to separate the results into high and low. Only 35% of these root stars

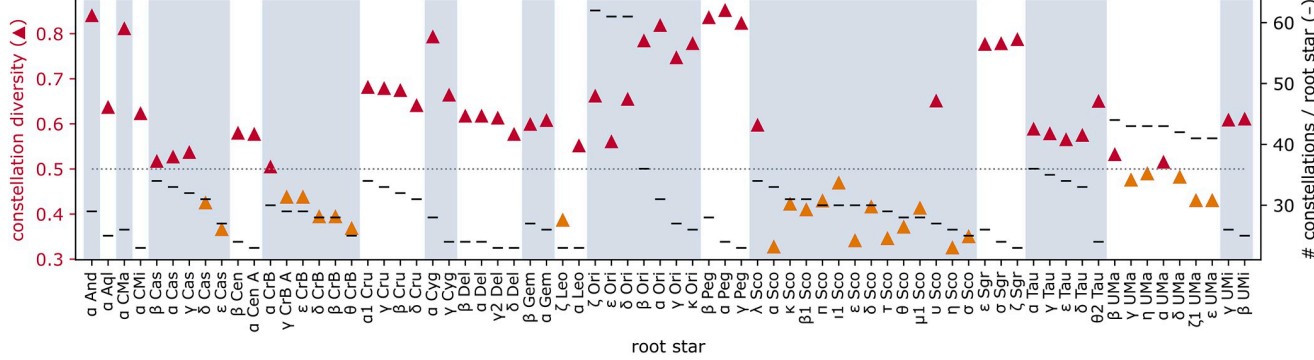

**Fig 9. Diversity by root star (II).** Per root star, the diversity index $H$ among all constellations with that star (on the **left** y axis with triangles), and the number of constellations with that star (on the **right** y axis with dashes). Value 0.5 for diversity is marked with a dotted line, and the diversity markers are coloured differently above and below this. The mean diversity is $H = 0.568$. The alternating background emphasises root stars from the same IAU constellation.

have a diversity index below 0.5, and there is no strong relationship between the number of constellations per star and the diversity index.

**A minority of root stars yield universal line figures**. Root stars in IAU Cas, CrB, and Sco score lowest. Constellations which include $\eta$ Sco (which lies on the tail half of IAU Sco) have the lowest diversity ($H = 0.325$). These line figures can only be found in two clusters: $\mathbf{C_2}$ for chain-like variants of the line figure, and $\mathbf{C_4}$ for variants with chains and added cycles. Constellations which include $\theta$ CrB (which forms one end of the IAU CrB chain) also have low diversity ($H = 0.367$) and can mostly be found in two clusters: $\mathbf{C_2}$ for chain-like variants of the line figure without cycles, like the IAU line figure, and $\mathbf{C_6}$ for variants in which the chain of stars is closed into a large cycle.

This low diversity can hypothetically be explained: characteristic to the sky regions in IAU Cas, CrB, and Sco is a chain-like pattern of bright stars, which then tends to produce relatively universal lines. The scorpion (and other long-tailed animals, such as snakes or stingrays) using stars in IAU Sco are indeed encountered in cultures which are geographically apart, and where the constellations were not Western-influenced: the Aztec, Maya, and Huave in Mesoamerica, Kari'na in S America, Maricopa in N America, Manus and Mandar in Oceania. Our results imply that this universality of the constellation shape is even broader than the recurring constellation semantics: line figures with long chains of stars are drawn around $\eta$ Sco regardless of what the constellation represents.

**The majority of root stars yield diverse line figures**. 65% of root stars have high diversity, and the peaks are in IAU And, CMa, Ori, Peg, and Sgr ($H > 0.8$). We provide one detailed example: the popular root star $\alpha$ Ori (one of IAU Orion's shoulders) has $H = 0.818$. The 31 constellations over this star span all clusters, although not equally. Fig 10 provides $\alpha$-Ori constellations as examples, to scale, with their embedding. Not all could be included; another example is in Fig 5: Wintermaker (Ojibwe). The high diversity is not only in shape, but also in constellation semantics (although not the subject of this study); many of the shapes do not depict a human figure, as in Mesopotamian and Greek traditions.

We conclude that a diversity of constellation designs appears more likely among popular sky regions. Exceptions occur for sky regions with special characteristics, such as a chain-like star pattern of bright stars, which produce universal line designs.

## 3 Discussion and conclusion

### Implications of the results

The map of constellations can serve as a *global taxonomy for line figures*, with its distinct clusters of single-link figures and chain-like figures characteristic to Eastern cultures, triangular meshes, non-planar spatial graphs, large loops, or the polygon-and-tendril designs characteristic to Western cultures.

The similarity graphs among cultures and culture types show that individual sky cultures do not have a unique visual signature, but cluster in three groups. The largest and most surprising is *a group of folk astronomies with oral traditions*, covering all continents. Culture *similarity aligns with ancestry* only for the region of influence of China and India, and for the root cultures of Western astronomy. In all other cases, *similarity does not align with ancestry*: other phylogenies have dense similarity links despite the geographical distance and lack of cross-influence. These findings imply that visual shapes over bright stars, using cycles, tendrils, and combinations, may be a natural, universal preference outside the zone of Chinese influence.

We found relatively *universal line designs in a minority of sky regions* around popular stars in IAU Cas, CrB, and Sco. Long chains of stars are drawn in these sky regions regardless of what the constellation represents: these shapes appear also when the constellation does not

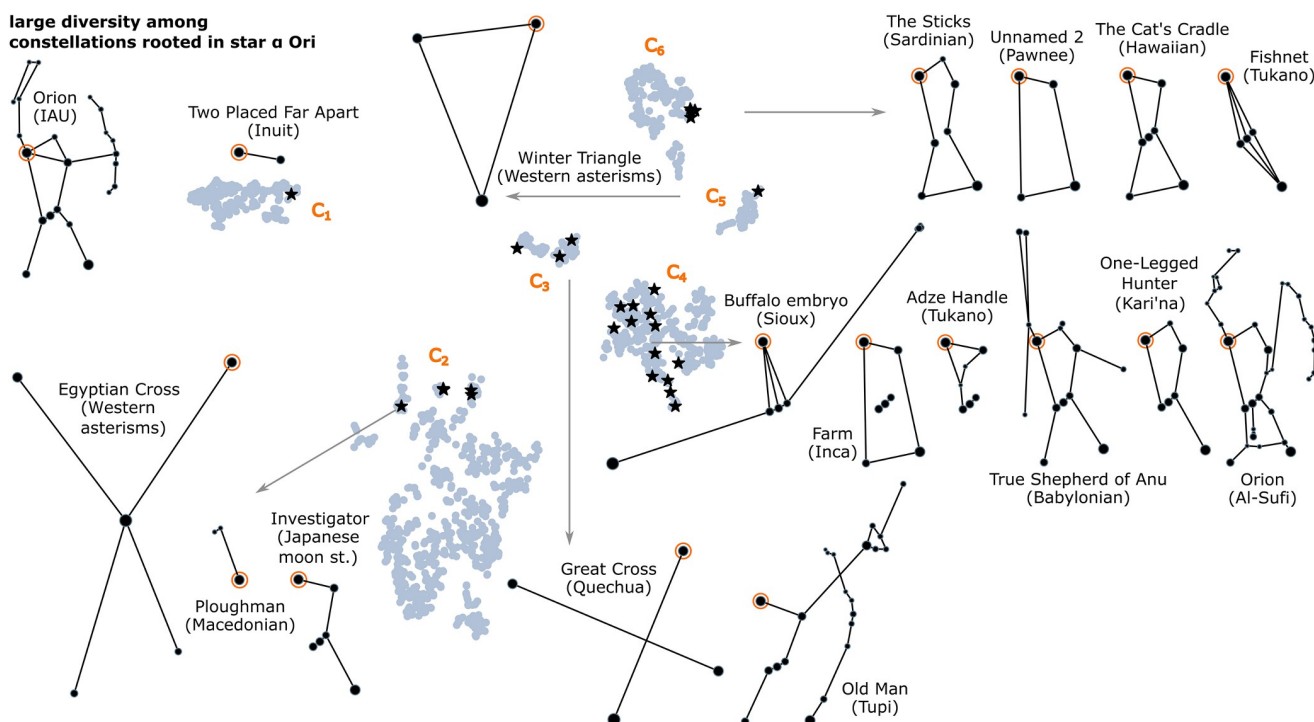

**Fig 10. The diversity of constellations over the root star $\alpha$ Ori (II).** In the background, in light blue, the embedding of all constellations. The IAU constellation Orion is shown at the top left, with $\alpha$ Ori emphasised. In the foreground, there are examples: black stars mark all constellations over $\alpha$ Ori, of which some are also shown, to scale.

represent the scorpion which recurs across cultures in the IAU Sco region. This implies that geometry is even more universal than semantics in these sky regions. On the other hand, we found *diverse line designs in a majority of sky regions*, which implies that diversity is the norm rather than the exception, and that the star pattern in a sky region determines the amount of diversity possible in line design.

## Limitations of the method

We assembled a diverse dataset of line figures from old and new astronomies around the world, but most oral cultures have an *incomplete record*, with their astronomy supplanted by modern tools, and possibly biased recall of the old traditions. Also, the *complexity of star constellations* is a debatable concept. Instead of fixing it into one dimension, we worked with a multidimensional definition, composed of many morphological features, which allows one to call different constellations complex for difference reasons. The features are interpretable, and the map of constellations in two dimensions retains this interpretability, with the feature gradient overlaid (as in Fig 4). However, the *choice of morphological features* for constellations also has limitations: it captures the spatial-network properties, but may consider two constellations dissimilar when human perception would not. For example, adding a link to close a cycle in a constellation makes a difference in network structure, but, if the link is very short, only a small difference in perception.

## Future work

This study provided global statistics across cultures, but a more focused study would be possible, by retaining part of the data and re-parametrising the embedding to separate smaller

subclusters. Also, the taxonomy of constellations, the similarity graphs for cultures, and the diversity results can help to guide future work which aims to find the cognitive principles of drawing constellation lines, or the features of the star patterns which yield specific constellation geometries.

## 4 Data: Inclusion, limitations, types, and timeline

**Overview**: The sky cultures in this study are listed in Table 1. The most difficult aspect of this research was collecting and verifying existing constellation data from a multitude of ethnographic and other sources, and categorising both cultures and constellations by type. We share the data at https://github.com/doinab/constellation-lines.

Column **location** in Table 1 locates the culture by continent. Column **id.** provides an identifier to important cultures: IAU (I), Babylonia (M from Mesopotamia), India (In), China medieval (C). Other identifiers (not defined in column **id.**) include the Greek (G), which comprises constellations in the standard IAU culture and is partly of Mesopotamian (M) ancestry, leading to indirect M ancestry for its descendants. Others denote cultural or migration regions: Austronesia (A), Polynesia (P), North America (nA) (with regional sky cultures from the Arctic to the south west, the Great Plains [14] and Mesoamerica), and South America (sA) (with two regional Guiana cultures, Kari'na and Lokono, two co-located Peru cultures separated only by time, Inca and Quechua, and two cultures now in Brazil, Tukano and Tupi—since some tribes migrated across this continent [12]). With these identifiers, column **an.** marks the ancestry or *phylogeny* of cultures. For a more detailed account of the phylogeny of sky cultures, see Sec. 4.2.

For each culture, column **type** marks two properties of the culture: whether astronomical knowledge transmission was written (w) or oral (o), and what the main practical uses were (at the time when the constellations were designed): for navigation (nv), religious (re) or political (po) divination, or folk/agrarian/hunter-gatherer time-keeping or orientation on land (fo). For details of this typology, see also Sec. 4.2.

Column **timestamp** provides the earliest date of a constellation record, and **count** provides the number of constellations with at least one line. In some cases, the first source for the sky culture is an ancient artefact, which only provides constellation names and some account of their location in the sky. Later work was needed to identify the stars and lines—the **references** in Table 1 point to modern data sources. These are of many types: short-form publications from field work or recent studies of medieval codices and other artefacts, books summarising constellation information from prior publications, language dictionaries which also document constellations, and sky cultures embedded in the Stellarium astronomy software [4]. Sec. 4.3 provides a timeline of line-figure representations for constellations, across cultures.

### 4.1 The criteria for inclusion, and limitations of the data

Not all cultures with documented astronomies, nor all constellations from a culture were included. The **inclusion criteria** are:

**Constellations have lines**. From the entirety of a sky culture, we study constellations with at least one line. This excludes constellations which are either single stars (frequent in E-Asian cultures), or tight star clusters such as the Pleiades, where lines would be invisible to an observer on the ground.

**Line figures are described**. The line figures come from literature, and are not a contribution here. Sources in which no line figures were drawn or described are excluded; this is the case

for many N-American tribes [14] other than those included in Table 1, the Tuareg [69], the Rapanui [70], and many others.

**Line figures are justified**. We exclude unreferenced Stellarium [4] data, with the exceptions: Western and Western asterisms (simple variants of IAU issued by popular magazines), Korea (from publicly available star charts), and Mongolia, Sardinia, Hawaii, and Vanuatu, which are recent, informally published field work with or by locals. Sometimes, a reference exists, but it only contains the star identification, pictographs, star groups, or a textual description of its shape; the line figure is tenuous, but follows the perimeter of the pictograph and the stars identified, so captures many of the constellation features (network size, spatial size, and brightness properties)—so are retained. This is the case for: Al-Sufi, India/Japan moon stations, Maya, and Inuit.

Of the 56 sky cultures, 27 come from ethnographic studies not already in Stellarium, so were gathered for this research. Of these, 3 (Sioux, Tukano and Tupi) are a mix of new literature and a prior Stellarium dataset. The Stellarium datasets were verified against their sources in most cases, and line figures were occasionally corrected or removed. Very faint stars unlikely to be seen with the naked eye (magnitude above 7.0), very rarely used, were removed and the line figures reconnected.

Not every type of analysis can be performed on this dataset, due to its **limitations**:

**Approximate star identification**. A star may be imprecisely identified from rough sky charts, in the neighbourhood of the star intended. This is especially the case for E Asian (Chinese, Korean, Japanese) sky cultures, for the non-determinative (or minor) stars in an asterism [71]. This means that the exact identity of a star in a line figure cannot be counted on.

**Unknown culture age**. The exact age of most sky cultures is unknown: their time of birth is lost in long periods without surviving records. This is the case for ancient cultures with written records, but also for recent cultures with oral traditions. Thus, age cannot be used as a culture feature, and questions cannot be asked about the evolution of sky cultures in time, or about the design of constellations in specific sky regions, such as around the poles (since the location of the celestial poles drifts due to the natural precession of the earth's axis).

**Unknown culture size**. The original number of constellations cannot be established for oral cultures, for many reasons. Part of the oral knowledge was not documented before becoming lost. There may have been taboos about sharing this information with strangers [14]. Some ethnographers did not have astronomical knowledge, so did not inquire about celestial traditions, or were vague about the identification of constellations in the sky. The result is poor ethnographic records for N America [14, 52], S America [18], Europe [38, 45], and the Pacific [62]. Thus, conclusions about cultures should not be drawn based on their sizes.

**Preference or bias towards bright stars**. In oral cultures, the constellations that were remembered were not a random sample from a larger, forgotten tradition. Instead, the recollection may have been *biased* towards the more salient constellations, such as those with bright stars. This is not certain; in some oral cultures, bright stars have utilitarian *preference*: in Polynesian seafaring, only they are named and considered major [59].

## 4.2 Sky cultures, their type, and phylogeny

Each paragraph introduces a related group of astronomical cultures from Table 1, whether the culture had written or oral traditions, their practical use (column **type** in the table), limitations in their documentation, and the ancestry (column **an.** in the table). Each culture or group of

cultures is emphasised in bold in the text when first mentioned. This is a too broad a subject to cover in depth here, so this general summary remains brief.

**Primitive astronomies: Moon stations**. The moon stations, also known as *lunar lodges* or *lunar mansions*, were groups of stars on the ecliptic. Between 27 and 28 moon stations (*nakṣatras*) or lunar mansions were documented during the **Indian** Vedic period (before 500 BC); if there was external influence in this early period, the literature is too late to provide information [31]. **Arabic** asterisms for time-keeping and orientation (the *anwā'*) also predate the Arabs' knowledge of other astronomies. At an unknown time, the Arabs received the Indian system of 28 lunar mansions [72], and each mansion was then identified with one of the Arabic *anwā'*. After the spread of the Greek astronomy, these asterisms were documented, in the 9th-century *Book of Anwā'* by Ibn Qutaybah [29]. Both systems were used for divination [29].

The 28 **Chinese** *xiu* (lunar lodges) were named ˜433 BC on a chest discovered in a tomb; they are likely much older (as old as the 3rd millennium BC), but evidence is lacking [73]. The question whether they were influenced by the Indian *nakṣatras* remains open [74]. **Japanese** astronomy closely followed the Chinese tradition. The Japanese *sei shuku* (lunar lodges) are based on their earliest sky chart, on the ceiling of a tomb from ˜700 AD, identified with the help of a later 17th-century chart [37]. The Indian, Arabic, and Japanese moon stations are recorded in this study in individual datasets; the Chinese moon stations are instead part of two Chinese sky-wide datasets.

**Early astronomies: Mesopotamia and Egypt**. The **Babylonian** sky is reconstructed from *MUL.APIN* [8], a clay-tablet compilation of Babylonian star catalogues produced up until that time. These were developed in stages from ˜3200 BC in two overlapping traditions, to represent gods and their symbols (the twelve signs of the zodiac and associated animals), and rustic activities. "Many constellations belonged to both traditions, but only the divine were transmitted to the West" [8]. We mark this culture as having primarily religious usage. The line figures follow pictographic representations of the zodiacal signs: on cylinder seals (from ˜3200 BC onwards), boundary stones (˜1350–1000 BC), in the Seleucid zodiac (clay tablets, with some copies surviving from the last few centuries BC) and the circular zodiac at the temple of Hathor in Dendera (an Egyptian ceiling bas-relief, ˜50 BC, merging Mesopotamian and Egyptian constellations) [8].

For the reconstruction of the ancient **Egyptian** sky, also with religious use [26], two pictographic references are used: the astronomical ceiling of the tomb of Senenmut at Deir el Bahari in Luxor (˜1470 BC), and the Egyptian figures on the Dendera zodiac [28]. In classical times, these native Egyptian constellations were combined with the Mesopotamian, producing the standard sky of the Greco-Roman period. We only use the older, native Egyptian constellations.

**Early astronomies: The Mediterranean**. The classical **Greek** sky (not a culture in this study, but included in all Western cultures) had 48 constellations and derived from two pre-Greek cultures [8, 9]. In 500 BC, the Greeks adopted the twelve Mesopotamian signs of the zodiac and associated animal constellations. These are, from oldest to newest: Taurus, Leo, Scorpius, and Aquarius (3000 BC, when they marked the cardinal points), Gemini, Virgo, Sagittarius, Pisces, and Capricornus (3rd or 2nd millennium BC). Aries, Cancer and Libra, the least bright, were accepted late, in classical times [9]. Other large constellations around the pole and equator of that time date from ˜2800 BC, probably originate from the Mediterranean region, and were designed as markers for sea navigation [9]. The Greek classical constellations of Mediterranean origin are bears (Ursa Major/Minor), serpents (Draco, Hydra, Serpens and Cetus), giants (Hercules, Ophiuchus, Boötes, Auriga), and some large southern marine constellations (Eridanus, representing a river meandering southwards). Likely, this set includes also Ara, Centaurus, Argo Navis, and Lupus [9]. The Greeks assembled these traditions 540-

370 BC, with the definitive documentation by Ptolemy in the *Almagest*, 150 AD. The **Al-Sufi Book of Fixed Stars** [30] is a revision of the *Almagest* with corrections and the addition of indigenous Arabic astronomical traditions.

All sky cultures of Greek ancestry have this mix of constellations of (originally) religious and navigational use. The practical use is thus per constellation, not per culture. The constellations with religious use are the Mesopotamian zodiac and their associates, Orion, and constellations associated with Greek mythology and their associates. Greek mythology characters and their vassals were: Cepheus, Cassiopeia, Andromeda, Perseus, Pegasus, Coma Berenices, Corona Australis/Borealis, Canes Venatici, Delphinus, Lyra, Canis Major/Minor, Lepus, Crater, Sagitta, Triangulum [9, 26]. Those with navigational use are the Mediterranean constellations, and, for more recent cultures, the constellations surrounding the south pole, attributed to various navigators [27]. Western constellations without a clear purpose when formed are left uncategorised.

**Early astronomies: East Asia**. Chinese astronomy is well documented; refer to [71, 73] for an extended description, of which we provide a summary. Chinese astronomy developed independently of external influences until as late as the 17th century. Some star groups were mentioned in East Asia before 1000 BC: the Big Dipper asterism on rock carvings, the name *Dou* (the Ladle, presumably for the Big Dipper) on bone inscriptions, and four asterisms including the Ladle, explicitly described, in folk songs. Qualitative descriptions of about 100 *xingguans* (asterisms) were given in a catalogue from the 1st century BC, and later catalogues list 280 asterisms. The only surviving celestial map until the 10th century is the Dunhuang star chart (a manuscript on paper dated around 700 AD), which draws the entire night sky visible from China (1345 stars in 257 asterisms). Astronomer Su Song's printed star chart from the Song period (1092 AD, with 1,464 stars in 283 constellations, which have become standard) is the oldest printed chart to survive. The Chinese asterisms are small, and name terrestrial items (the imperial family, officials, domestic animals, crops, and buildings). They were used for astrological predictions at the imperial court [71].

The **China medieval** dataset draws the Song-period chart, *Xinyi xiang fayao* (New design for an armillary and globe), using books of accurate stellar measurements by officials from 1052 AD; this information was compiled in a book [3]. The later **China** dataset draws the skies based on a star catalogue revised by Qing-period officials with the help of Western astronomers, finished in 1756 [3], and later additions to this. The Western astronomers did not supplant the Chinese asterisms with Western ones, but measured star positions more precisely, occasionally added stars to asterisms, and added stars near the south pole which were invisible from China. The **Korean** dataset follows the earliest surviving Korean chart, a marble stele dated 1395 AD, *Ch'onsang yolch'a punyajido* (Chart of the regular division of the celestial bodies). This chart is a reproduction of an older Chinese chart, although some line figures differ. Modern measurements of the star positions suggest a date around 30 BC—so this may preserve traditions older than the surviving Chinese charts [73]. The identification of the stars on these maps is tenuous, due to the irregular projections and imprecise star placement.

**Western sky cultures**. The International Astronomical Union (IAU) standardised a list of 88 constellations with names and three-letter abbreviations (in 1922) based on the classical Greek sky and later discoveries in the southern hemisphere. At that time, the constellation figures had vague and variable perimeters. Standard constellation boundaries were published in 1930 [75]. Line figures had been sketched by French astronomers, Alexandre **Ruelle** and Charles **Dien**, on the first modern-looking star charts with line figures and without pictographs [39, 40] (dated 1786 and 1831), including constellations which are now obsolete. We transcribe these early charts in two datasets. The first popular line figures were H. A. **Rey**'s [10] (1952) intuitive figures of the objects they are supposed to represent. They largely adhere to previous traditions, but also sometimes deviate from the figures described since the *Almagest*.

For example, Rey's bear in Ursa Major is oriented the opposite way compared to Ptolemy's description. The **IAU** line figures, "Alan MacRobert's constellation patterns [from the Sky & Telescope magazine] were influenced by those of H. A. Rey but in many cases were adjusted to preserve earlier traditions" [11]. The **Western** constellation set is a simpler variant used by the popular astronomical software Stellarium [4]. Separately, we have the set of popular **Western asterisms** which do not respect IAU constellation lines, such as the Spring and Summer Triangles (with various sources provided for the figures [4]); we mark this is a folk culture.

The **Belarusian** folk sky is compiled by a local ethnoastronomer [41] from sources in the 19th to 21st centuries. Most lines are a subset of the IAU constellations, with different names and meanings. The **Romanian** sky is the result of an early study (1896) by a local mathematician and teacher [42], who sent copies of a sky map to teachers throughout Romania, requesting them to ask the oldest peasants about their beliefs about constellations. In 39 of the papers returned, there were new accounts of constellations; this makes the collection unique in depth among the few documented astromythologies of Europe. **Macedonian** ethnoastronomical research began in 1982, with 140 villages visited for surveys. Given the symbolism of the constellations, the roots of this sky culture are likely in early agricultural communities of the Neolithic in the Balkans [44]. The **Sardinian** data is the result of unpublished research by locals [4]. Only four constellation shapes are identified well across **Russian**-speaking republics from the early 20th century onwards; the collectors of Russian folklore were not familiar with the sky, leading to poor ethnographic records [38].

The present **Mongolian** constellations are of Western influence (although the Mongolian mythology has its own identity [4], and earlier cultures may have been influenced by the Chinese). Knowledge transmission in this culture remains essentially oral; this data was collected in 2014 from field work.

All these folk cultures had oral traditions, a mostly agrarian iconography, and influence from the IAU constellations.

**Norse** (particularly Icelandic) astronomy may have been well developed for navigation and time-keeping, but little has been preserved. Greek and Latin names for constellations supplanted local ones in the medieval period, and continental influence is likely. The only old sources are literary: the Eddas (from 13th century Iceland), and a compilation of timetelling verse [45] give Norse interpretations to parts of modern constellations. The **Sami** sky culture may be very old, but written records for this oral tradition of reindeer herders exist only since the 19th century [46]. There are few constellations, all connected to Sarva the Elk. The Sami sky culture doesn't resemble the Western, and there is no documented influence.

**North American sky cultures**. In Mesoamerica, the ancient **Maya** sky is due to research [47, 48] based on imprecise information, namely the partial pictographs of animal constellations in the *Paris Codex*, dated around 1450. Unlike the Maya sky, the **Aztec** sky is based on line figures and textual descriptions from 16th-century codices (line figures in *Primeros Memoriales* [49], and descriptions in the *Florentine Codex* [50]; the challenge was in locating the stars and constellations [51], which is approximate. Both Maya and Aztec cultures kept written astronomical records, which served a divinatory function. The current constellations of the **Huave** in Mexico are only used for telling time, and retain some pre-colonial influences.

The **Native American** sky cultures outside Mesoamerica are all oral traditions documented recently, between the late 19th century and the present, with a practical use as time-telling tools for farming, hunting, and gathering [76]. Their traditions retained some originality, with star mythology related to hunting and planting [14]. There are occasionally striking commonalities: natives in the Northeast, Southeast, western Subarctic, and the Plateau all have myths about the Big Dipper asterism in Ursa Major as a bear and hunters. The datasets are small due to lost traditions and a lack of timely documentation [14]. "Many of the [Inuit] elders insisted

that the information they possessed about astronomy was meagre compared to that of their parents or grandparents; the present generation of elders is the last repository of a more or less detailed knowledge on this subject" [53].

**South American sky cultures**. The traditions are oral and represent the natural world. Like the Maya sky, the coastal **Tupi** have constellations documented in an 1614 book [57], only later identified by 20th-century studies via comparisons with the constellations of other tribes [12, 58]. Due to migrations, there is influence between the coastal Tupi, Amazonian, Guianan and Andean tribes [12]. In the Andes, the **Inca** constellations are a 1928 interpretation [54] of a 1613 astronomical drawing in Cuzco, Peru; the identification uses textual descriptions from co-located tribes, Aymara and Quechua. Field work from 1978 also documents the little that remains of these constellations of the **Quechua** [55], now heavily Spanish-influenced. On the coast, the **Lokono** territory borders that of the **Kari'na**; though unrelated, they share a Guianan culture, and the ethnoastronomical traditions have similarities. The Lokono constellations are recent ethnographic work [18]. Those of the neighbouring Kari'na were documented from before 1900 to 1980 (the latter, field research in three Carib villages from Suriname [13]). The Kari'na sky traditions are fading fast: many of the constellations which were mentioned in a 1907 study were unknown by 1980. For the Amazon, **Tukano** constellation data was provided in research reported in 1905 [56] and 2007 [17], with all figures in a similar style. We add them up as the Tukano sky culture.

**Polynesian sky cultures**. Stars and constellations are the most important tool for long-range oceanic navigation for the Polynesians (Hawaii, Maori, Tonga), outlying islands which are culturally Polynesian (Anuta, Vanuatu), and others (Caroline, Marshall, Kiribati, Manus), all with ancestry in a common seafaring culture. Their knowledge was transmitted orally. Some constellations were still known by **Anuta** sailors at the end of the 20th century, and were identified and then drawn during field research in 1972-83 [59]. The **Hawaii** "star lines" were lost by the 1970s, then reconstructed with help from a Micronesian navigator [62]: a star line is a group of main stars connected into lines and simple shapes, pointing to main cardinal points. **Tonga** data is a synthesis of the sailing directions written by a high Tongan chief in the late 19th century [67]. **Maori** data is a synthesis of 19th-century information and fieldwork [65]. The **Vanuatu** data comes from field work with locals on the Tanna island by an amateur astronomer in 2019, published informally; the researcher also states that "in many places of Vanuatu this ancestral knowledge is now almost forgotten" [68]. This data is an outlier for this geographical region: all constellations represent agricultural concepts. The **Marshall** and **Kiribati** data was digitised from language dictionaries which include names and descriptions for constellations, initially written in the 1970s, and updated since [63, 66]. The data for the **Carolines** comes from field work published by 1951 [60] with a later correction of names [61]; that for **Manus** is a summary of field work done throughout the 20th century [64].

**Austronesian sky cultures**. All four are oral cultures, recently documented. The **Mandar** [32] and the **Bugis** [32, 33] are two neighbouring ethnic groups on the island of Sulawesi in Indonesia. Both have a strong seafaring tradition due to the Austronesians which migrated into Sulawesi. The Mandar constellations are fading from memory: the data was collected "through interviews with retired fishermen. We noted that the younger fishermen did not use star patterns for their navigation" [32]. The Bugis data is a unified drawing of constellations from prior sources [33]. On the island of **Madura**, Indonesia, off the coast of Java, the locals also have a maritime tradition, likely also of Austronesian origin. The constellation data was collected via a survey of about 100 families [36]. Nearby on the island of Java, the **Java** astronomy serves an agrarian society, cultivating rice. Information for one constellation (The Plough in IAU Orion) was first provided in person to the author of a 1885 study [34] by a religious figure from an English mission on the island. This was confirmed and supplemented with two

others in a recent paper [35] with data from 2019 interviews; the authors also confirm "widespread similar asterisms in the archipelago".

## 4.3 Line figures: Both ancient and modern representation for constellations

The line-figure representation is new in many cultures, but native in others. We summarise the timeline (also in Fig 11) and the sources for line figures, emphasising any limitations in their collection.

**From ancient pictographs or text to recent line-figure interpretations**. The ancient skies in **Babylonia** and **Egypt** are based on pictographs, without clearly identifying the stars, so the line-figure identification is tenuous. The reconstruction differs among researchers (for both Babylonia [7, 8] and Egypt [28]). We use the most recent and extensive identifications [5–7, 28]. The line-figure identification is tenuous also for the **Inca**, **Maya**, and **Tupi** skies, for which the early 15th- and 17th-century sources do not identify stars and rarely draw line figures: the Inca source draws the Stove constellation (equivalent to IAU Crux) with lines, but the rest are from recent identification research.

Also of ancient pictographic origin, the sources of IAU constellations do identify most stars, but never had standard line figures. There are instead many early and current variants based on the *Almagest* [77]), such as **Al-Sufi**: the *Book of Fixed Stars* [30], like *Almagest*, draws pictograms and places the stars with their astronomical data (latitude, longitude, magnitude) in the pictogram ($\alpha$ UMi is "the star at the end of the tail" [77]).

The **Norse** constellation Aurvandil's Toe is mentioned in the Eddas (only in verse), and its identification with IAU Corona Borealis is likely, but not certain [4]. The remaining Norse constellations use simple line figures [4, 45]. The **Indian** [31] and **Arabic** [29] moon stations consisted of well-identified stars, without lines; the dataset draws very basic lines to link these stars (in most cases a chain).

**From ancient oral traditions to recent line-figure interpretations**. The traditions of native tribes in **N America** outside Mesoamerica had very few artifacts recording star groups (without lines): pictographs on the ceilings of Navajo caves show constellations from the 1700s; early 1900s Navajo and Tipai sandpaintings and an undated Pawnee star chart on buckskin show recognisable star groupings [14, 78]. The line figures are interpretations of the researcher. The same is the case for **Lokono** in S America: the researchers write that "many Lokono, when drawing constellations, did not connect the stars with lines and that in some cases, there was little agreement among speakers as to which star within a constellation corresponds to which parts of the plant or animal it represents; the Lokono tradition allows for more flexibility in interpreting particular star groups" [18]. The **Tukano** also drew their constellations in the form of star clusters without lines; it is the researchers who added the lines: Koch-Grünberg as a printed star chart in his 1905 book [56], and a dataset based on a 2007 dissertation [17]. This process was similar for the **Quechua** [55].

Other native tribes did provide line figures. The **Huave** in Mesoamerica provided constellations which clearly resemble certain animals or religious objects in contour, so the shape of the figure is certain. The **Kari'na** line figures in S America "have been drawn in the sky by villagers and referred to Western sky-maps by us" [13]. The Indonesian (**Mandar**, **Bugis**, **Madura**, **Java**), and culturally Polynesian (**Anuta**, **Vanuatu**) constellations were identified with locals and drawn by the researchers as either pictograms or line figures [33–36, 36, 59, 68]. The oldest explicitly drawn line figure is from an 1885 Javanese source [34]. When pictograms are drawn, they are a contour (or lines through) the stars identified, so are fairly certain.

Other Polynesian (**Maori**, **Tonga**), neighbouring Micronesian (**Marshall**, **Kiribati**, **Carolines**) and Melanesian (**Manus**) constellations were described only in text [60, 63, 65–67], but

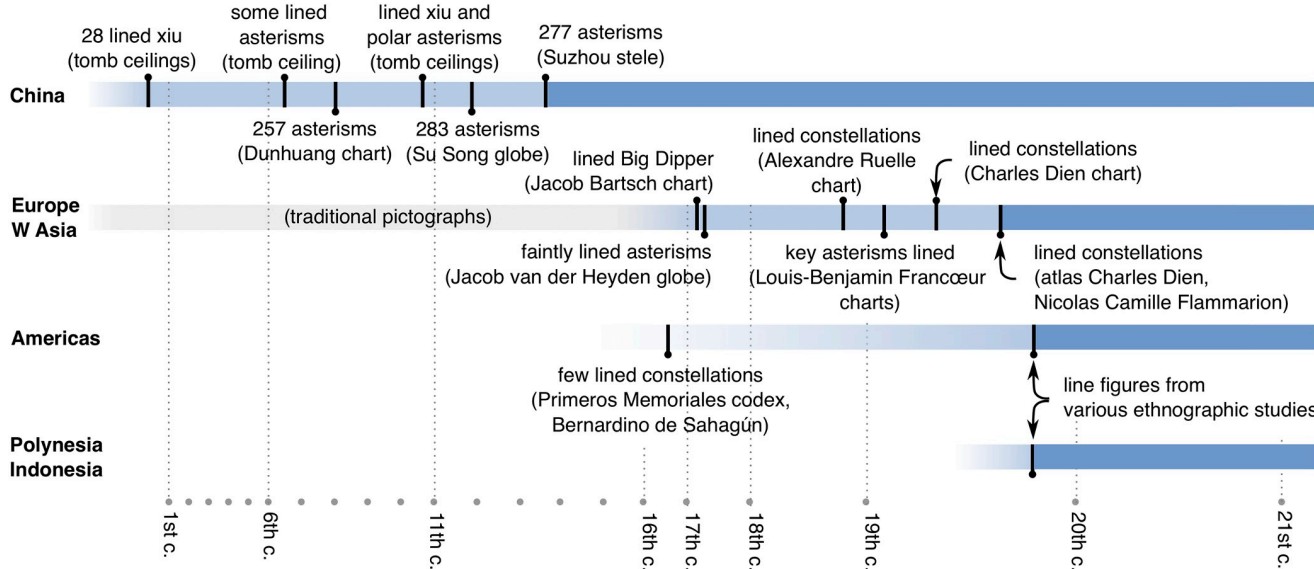

**Fig 11. Regional timeline for line figures.** The centuries AD are marked at the bottom, on a non-linear scale. Dark blue time intervals are recent periods, during which line figures are documented. They are preceded by periods of development or transition (in light blue), during which important sources for line figures are marked. In E Asia, the circle-and-line representation is native. In Europe and W Asia, line figures developed out of pictographs, alongside the development of modern astronomy. In the Americas and the Pacific, they were likely introduced under Western influence.

simple lines correspond well to this text. For example, "Toloa means a wild duck, and Tongan imagination pictures the cross as a duck whose head is $\gamma$ and tail $\alpha$, the wings being $\beta$ and $\delta$ [Cru]" [67]). The Marshallese constellation "Aoḻōt is shaped like leather-jack fish ($\theta$, $\eta$, $\zeta$ Dra) whose head has been pierced by a spear (18, 19 Dra)". In the Carolines, for IAU Cru, sources confirm that the figure follows the name, so can be drawn with certainty: "the Micronesians do not see a cross, but rather a triggerfish with its distinctive diamond shape" [79].

Some oral sky cultures in Eurasia (**Russia**, **Sami**, **Belarus**, **Macedonia**, **Romania**, **Sardinia**, **Mongolia**) were documented recently (in the 19th century or later, by researchers using sky charts), and described all line figures at collection time. Most researchers provide line drawings directly. Others describe them in words, such as the Belarusian report [41] (constellation The Cross is "$\alpha$, $\gamma$, $\eta$, $\beta$ Cygnus—a vertical beam, $\epsilon$, $\gamma$, $\delta$ Cygnus—a cross beam").

**Native line figures of all ages**. In E Asia, line figures were and remain the only standard representations for star asterisms; on a **Chinese** tomb ceiling (uncertain date BC), the moon stations are depicted with stars as small circles, joined into groups by straight lines. The Dunhuang star chart and its successors in China, Korea, and Japan, draw line figures, with the stars of roughly equal size regardless of magnitude, annotated with star names. In the Americas, the *Primeros Memoriales* codex [49] drew line figures for the **Aztec** sky in the 16th-century, and one **Inca** constellation is lined in the 17th-century source [54]. In the Western world, the first complete star charts with line figures were by **Ruelle** [39] (1786) and **Dien** [40] (1831). **Rey**'s line figures [10] (1952) became widely adopted, followed by variants, such as the **IAU** and Sky & Telescope magazine's [11] and the simpler **Western** sets [4]. All are included in this study.

## 5 Method: Measuring, clustering, and comparing visual signatures

To provide answers to the questions posed, the steps followed are: measuring the constellation visual signatures (described here in Sec. 5.1), clustering constellations and interpreting the

clusters (Sec. 5.2), and finally measuring the association of culture, type, or sky region to the visual signature (Sec. 5.3 and 5.4).

## 5.1 Measuring the visual signature of a constellation

A constellation is a *spatial network* in the astronomical coordinate system. The *nodes* are stars annotated with locations on the celestial sphere (independent of the location of observers on the earth): the declination $\delta$ (equivalent to the geographic latitude) is the north or south angle between the celestial equator and the star; the right ascension $\alpha$ (equivalent to the geographic longitude) is the angle measured along the celestial equator. An additional annotation is the magnitude, a measure of the star's visible brightness. A *link* (or line) is the undirected geodesic between two nodes.

We define a set of 19 **constellation features** or statistics. For definitions of the network features ($s_1$ to $s_{11}$), refer to [80]. All features jointly form the visual signature of the constellation. We listed the features in Sec. 2.1. Here, we clarify the choice.

Statistic $s_1$ is the **number of links**, a measure of the network size, while $s_4$ is the **clustering coefficient** (the average of all local clustering coefficients), a measure of how close the network is to being a triangular mesh. $s_5$ is the **maximum core number** (the largest $k$ so that there is a maximal subgraph containing nodes of degree $k$), a measure of whether the network has one or more dense core regions, distinct from sparsely linked peripheral regions. $s_6$ (the **number of cycles**) and $s_7$ (the size of the **largest basic cycle**) are computed over the cycle basis (the minimal set of cycles so that any cycle in the network is a sum of these). $s_9$ and $s_{10}$ are the average **link diameter** and shortest path among the connected components (the latter shows whether the network is **small-world**). $s_{11}$ is the **link connectivity** (the minimum number of links which, if removed, disconnect the network). Other constellations statistics could be defined; this set was selected to cover fundamental visual aspects of the line figures, and to exclude redundant measures. For example, the number of nodes in the network is not explicitly included, but is proportional to $s_1/s_3$. These statistics also do not count on the stars being identified precisely (given the limitations of the data collection): if a star was mistaken for another in the close neighbourhood and of similar magnitude, the statistics change little.

We opt for this highly *multidimensional* definition for the visual signature, instead of defining a small number of more abstract or high-level measures of visual complexity, such as the perimetric complexity applicable to written characters [1] or binary images [81], and the algorithmic, representational, or algebraic complexity applicable to 3D shapes [82] or written characters [1]. Each of our features remain interpretable, are similar to other known measures of shape complexity, such as morphological and combinatorial complexity [82], and capture the salient characteristics of these spatial networks over scattered bright points on a sphere. This allows us to understand the word 'complexity' in more than one way: a spatially small ($s_{12}$) but sharp-angled ($s_{14}$) constellation with closed polygons ($s_6$, $s_7$) such as IAU Cygnus can be called complex for reasons different than a long ($s_9$), spatially large ($s_{12}$), but entirely linear ($s_2$) constellation such as IAU Eridanus.

## 5.2 Embedding visual signatures: Nearest-neighbour dimensionality reduction

We use *manifold learning* [83] out of the high-dimensional constellation features, to optimise the projection (or *embedding*) of this data into two dimensions. The objective is to preserve the neighbours of a constellation (by Euclidean distance, which captures the similarity across all features) between the original many dimensions and the final two dimensions. Constellations

which are *similar* in the original space remain neighbours in the embedded space, and can now be visualised and interpreted.

The *t-distributed Stochastic Neighbor Embedding* (t-SNE) [23, 24] is manifold learning which also separates any clusters in the data. Conclusions can thus be drawn from the embedding at multiple scales: some constellations will be local neighbours, while some clusters of constellations will be regional neighbours on a higher scale. T-SNE visually disentangles data that has internal structure at different scales. This embedding serves as a clustered summary of constellation visual signatures.

**Tuning the manifold learning**. We run the t-SNE [23, 24] implementation from scikit-learn [84] and parametrise it as in Table 2. We aim for two dimensions, which can be visualised clearly. The distance used is squared *Euclidean distance*, which is calculated after *scaling* the data feature-wise with a standard scaler, which removes the mean of the distribution and scales the feature to unit variance. First, t-SNE converts the closeness of data points into joint probabilities, and minimises their Kullback-Leibler (KL) divergence in the original vs. the embedded space. The gradient-descent algorithm does not guarantee that it will reach the optimum KL divergence, so we try different random seeds (that is, do multiple restarts) and select the best divergence found. The algorithm is computationally heavy—the runtime of manifold learning with 20,000 iterations is on the order of tens of minutes, even though the number of data points (constellations) is relatively small, in the thousands. Two identical constellations are embedded at roughly the same coordinates.

The *perplexity* parameter $p$ matters: it is essentially the number of nearest neighbours to consider. We set this to the average culture size (the number of constellations per culture, here 32). This makes regional structure visible. A smaller value for $p$ would be useful if splitting large clusters is desirable.

**Evaluating the embedding**. The *trustworthiness* metric [85, 86] expresses to what extent the local structure (the neighbourhood of all constellations) is retained by the embedding. Given a number of nearest neighbours as parameter (here, equal to the perplexity value from Table 2) and a type of distance (here, Euclidean), the trustworthiness is a value in [0, 1] which reaches 1 if all constellations have retained all their nearest neighbours after the embedding. We report the trustworthiness value to evaluate the embedding itself.

**Interpreting the embedding**. For t-SNE, the interpretation of the embedding has some caveats. The scale of the two embedded dimensions has no meaning (similar to the dimension of a distribution after standard scaling), so is never shown. The size of clusters in the embedding (for example, the area of the convex hull around a cluster) also has no meaning, because the t-SNE algorithm adapts to regional density variations in the data, and equalises these. The composition of the clusters is meaningful, although dependent on the perplexity parameter.

For both of our hypotheses, the predictors are categorical variables: the culture name, the type of knowledge transmission or type of practical use, the ancestor culture, or the sky region. The questions ask whether these categorical variables associate (and may have driven) the

**Table 2. Parameters for the t-distributed Stochastic Neighbor Embedding (t-SNE).**

| t-SNE parameter | value |
| --- | --- |
| embedding dimensions | 2 |
| initialisation | Principal Component Analysis |
| gradient calculation algorithm | exact |
| learning rate | 50 |
| max. number of iterations | 20,000 |
| perplexity $p$ | 32 |

numerical constellation features (defined in Sec. 5.1 above). For Hypothesis I, the next step is to measure the association between the categorical predictors and visual signatures.

## 5.3 Measuring association with culture types: Assortativity and similarity metrics

The scoring of questions **I.1-4** proceeds as follows. Out of the 19-dimensional dataset of constellation features, we compute a *directed, nearest-neighbour network of constellations*: for each constellation, outlinks are drawn to the $p$ nearest neighbours by Euclidean distance. We again set $p$ equal to the average culture size, 32. The nodes in this network are marked with the culture type; for example, for question **I.1**, each constellation is assigned the name of its culture.

**The assortativity coefficient**. This normalised score measures the degree of mixing, and is similar to the intra-class correlation coefficients used to compare different groups in a population [87]. Assortativity is also referred to as *homophily*. An $r = 0$ means that the cultures mix randomly, or that the constellations are equally similar intra-culture as they are inter-culture. A value $r > 0$ signals instead that the constellations are similar intra-culture and dissimilar inter-culture, so the culture is indeed a predictor of the visual signature. A negative value for $r$ is possible, and signals the opposite, that constellations are more similar inter- than intra-culture.

The assortativity coefficient $r$ is a classical (2003 [25]) measure for assortative mixing in a graph. We repeat the definition here for completeness; examples and a discussion are available in [25, 80]. Given a directed network whose nodes have a categorical type marked (denoted $i$ or $j$), a square *mixing matrix* can be formed out of this graph. This matrix has as many rows as there are distinct node types. A cell, denoted $e_{ij}$, contains the *fraction* of directed edges in this network which connect a vertex of type $i$ to one of type $j$. The sum of all cells in the mixing matrix is 1. The fraction of edges which have as source any node of type $i$ is denoted $a_i$, and likewise $b_i$ for destinations of type $i$. The expressions for $a_i$, $b_i$, the assortativity coefficient $r$, and the expected statistical error $\sigma_r$ are:

$$a_i = \sum_j e_{ij}, \quad b_j = \sum_i e_{ij}, \quad r = \frac{\sum_i e_{ii} - \sum_i a_i b_i}{1 - \sum_i a_i b_i}, \quad \sigma_r^2 = \sum_k (r_k - r)^2,$$

where $r_k$ is the value of $r$ for the network in which the $k$th edge is removed (a standard method using link removal [25]).

Intuitively, $r$ compares the diagonal of the mixing matrix (the total fraction of edges between nodes of the same type, regardless of the type) with the expected value for this total fraction, as given by random chance: $a_i b_i$ is the expected fraction of same-type edges, given the relative fraction of those types in the data. This definition is normalised, so the maximum value is 1. However, this maximum value of 1 is not reachable for any network, for example when the node outdegrees are larger than the number of nodes from one type [80]. When this occurs, we report $r$ normalised, divided by its empirical maximum value.

When one type of constellations is much less numerous than other types (since the cultures are heterogeneous in size), the assortativity coefficient treats any constellation equally, so less numerous types are not weighted more than more frequent ones [25]. This makes for a stable coefficient $r$, which changes little if one constellation or link is added or removed from the data.

We provide both the global assortativity (for example, for question **I.1**, a mixing score for all cultures), and also a one-versus-others assortativity, which measure how a particular predictor value mixes with all others taken as one group (for example, for question **I.4**, the mixing of all cultures of Chinese ancestry with all other cultures).

**The similarity metric**. When the results signal positive assortativity, we also measure the similarity between any two culture types. This similarity metric is a single statistic, defined in this work. Given two classes $c_1$, $c_2$ (for example, two cultures), we compute:

- $r$: over the constellation network consisting of three labels: $c_1$, $c_2$, and *other*;

- $r_m$: over the network with class labels $c_1$, $c_2$ merged into a single class $c$.

We define the *similarity*$\Delta$ as their signed difference, $\Delta = r_m - r$. $\Delta$ is positive if merging the two classes raises the assortativity of the network, or makes the mixing of classes *less* random; this signals similarity in the merged classes. $\Delta$ is negative if merging the classes makes the mixing of classes *more* random, so signals dissimilarity. We build a graph of similarities between the predictor values of each question, drawing links weighted by $\Delta$ between classes, when $\Delta$ is positive.

## 5.4 Measuring diversity per sky region

We define a sky region not by its boundary coordinates, but by a *root star*: a sky region is the *set of constellations* (from any culture) which all include a given root star. This definition is less biased than alternatives: by not limiting a region to certain celestial coordinates, we remove any assumptions over the size of constellation that may be drawn there. Sky regions defined by two neighbouring root stars will overlap.

For Hypothesis II, the problem is of a different nature than for Hypothesis I, and thus requires a different scoring. Here, we do not compare sets of constellations, but measure how *diverse* (or, on the contrary, uniform) the constellations of a single set are, over the clusters of constellations (described in Sec. 5.2). Assume that the embedding shows $k$ clusters of visual signatures, and the sky region under study is a set of $n$ constellations. Maximum diversity is achieved when the $n$ constellations are distributed equally over the $k$ clusters, and minimum diversity when all constellations fall within one cluster.

We use the classic Shannon diversity (or entropy) index $H$ [88], a common score for diversity. $H$ measures how much "choice" there is among the values of a discrete distribution. In this case, it measures how constellations from a given set distribute among discrete clusters of visual signatures. Assuming that the embedding shows $k$ clusters of visual signatures (indexed by $i$), and the sky region under study is a set of $n$ constellations, the diversity index is:

$$H = - \sum_i p_i \log p_i,$$

where $p_i$ is the *fraction* from the $n$ constellations which fall into cluster $i$. $H = 0$ if all constellations fall into one cluster. The maximum value, $\log k$, is reached if the constellations distribute equally among all clusters. We report $H$ normalised, divided by this maximum value.

## Acknowledgments

The author wishes to thank the Stellarium team (particularly Dr. Susanne M. Hoffmann and Fabien Chéreau), the contributors to the Stellarium repositories for sky cultures across the world, Prof. Charles Kemp, and Dr. Pieter-Tjerk de Boer.

## Author Contributions

**Conceptualization:** Doina Bucur.

**Data curation:** Doina Bucur.

**Formal analysis:** Doina Bucur.

**Investigation:** Doina Bucur.

**Methodology:** Doina Bucur.

**Project administration:** Doina Bucur.

**Resources:** Doina Bucur.

**Software:** Doina Bucur.

**Validation:** Doina Bucur.

**Visualization:** Doina Bucur.

**Writing – original draft:** Doina Bucur.

**Writing – review & editing:** Doina Bucur.

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
