## [Decision Letter · Decision Letter 0]

6 May 2022

PONE-D-21-36117The network signature of constellation line figuresPLOS ONE

Dear Dr. Bucur,

Thank you for submitting your manuscript to PLOS ONE. After careful consideration, we feel that it has merit but does not fully meet PLOS ONE’s publication criteria as it currently stands. Therefore, we invite you to submit a revised version of the manuscript that addresses the points raised during the review process.

We look forward to receiving your revised manuscript.

Kind regards,

Diego Raphael Amancio

Academic Editor

PLOS ONE

Journal Requirements:

2. We note that Figure 2  in your submission contain map images which may be copyrighted. All PLOS content is published under the Creative Commons Attribution License (CC BY 4.0), which means that the manuscript, images, and Supporting Information files will be freely available online, and any third party is permitted to access, download, copy, distribute, and use these materials in any way, even commercially, with proper attribution. For these reasons, we cannot publish previously copyrighted maps or satellite images created using proprietary data, such as Google software (Google Maps, Street View, and Earth). For more information, see our copyright guidelines: http://journals.plos.org/plosone/s/licenses-and-copyright.

a) You may seek permission from the original copyright holder of Figure 2 to publish the content specifically under the CC BY 4.0 license.  

Reviewers' comments:

Reviewer's Responses to Questions

**Comments to the Author**

1. Is the manuscript technically sound, and do the data support the conclusions?

Reviewer #1: Partly

Reviewer #2: Yes

2. Has the statistical analysis been performed appropriately and rigorously? 

Reviewer #1: I Don't Know

Reviewer #2: Yes

3. Have the authors made all data underlying the findings in their manuscript fully available?

Reviewer #1: Yes

Reviewer #2: Yes

4. Is the manuscript presented in an intelligible fashion and written in standard English?

Reviewer #1: Yes

Reviewer #2: Yes

5. Review Comments to the Author

Reviewer #1: The authors performed a massive study of star constellations in an attempt to correlate line figures with distinct cultures across human history. They explained their methodology - based on network science - and their results in great detail. Because they were very ambitious in their goals, which is good, the authors failed to provide a clear message to the readers. Unless the reader can make notes and pay great attention to the details and figures, he/she cannot infer which the main conclusions are. All one can learn is that constellation line figures can be classified according various criteria and the classification correlates poorly with cultures. Some distinction can be made between oral and written astronomies.

Therefore, a recommend a major revision for the manuscript to be useful in conveying a message to PLOS One readers, as follows.

1) The abstract should make it explicit how network science is applied to get the results. This piece of information is hidden.

2) The authors should select a few items for comparison in which there is either a high or no correlation, so that the conclusions could be more easily understandable by the reader.

3) The Conclusion section ends abruptly with description of methodology, and the reader is left without a comment about the implications of the results.

Reviewer #2: This is one of the most exciting papers I have read in the last 5 years. I would like to thank the author for the novel approach and study of an interesting topic. I personally learned a lot and would like to see this work published with some improvements.

I find the methodology solid in terms of feature extraction and projection into low dimensional space for visual analysis. My only concern is that almost all of the hypotheses are supported by the interpretation of the tSNE projections. I think the author can improve the approach by creating ML models and use measures of performance to support relevant research questions. For instance classification problem or applying clustering to the data and measuring accuracy using the labeled data (e.g., in Fig5).

In my opinion, the paper could also benefit from restructuring, as the background information and interpretations are very text-heavy and make it harder to follow. Perhaps the authors could focus on the most striking results first and then provide descriptive statistics of the analysis.

6. PLOS authors have the option to publish the peer review history of their article (what does this mean?). If published, this will include your full peer review and any attached files.

Reviewer #1: No

Reviewer #2: No

---

## [Author Response · Author response to Decision Letter 0]

23 Jun 2022

Responses to reviewers are in a 5-page .pdf letter.

---

## [Decision Letter · Decision Letter 1]

18 Jul 2022

The network signature of constellation line figures

PONE-D-21-36117R1

Dear Dr. Bucur,

We’re pleased to inform you that your manuscript has been judged scientifically suitable for publication and will be formally accepted for publication once it meets all outstanding technical requirements.

Kind regards,

Diego Raphael Amancio

Academic Editor

PLOS ONE

Additional Editor Comments (optional):

Reviewers' comments:

Reviewer's Responses to Questions

**Comments to the Author**

1. If the authors have adequately addressed your comments raised in a previous round of review and you feel that this manuscript is now acceptable for publication, you may indicate that here to bypass the “Comments to the Author” section, enter your conflict of interest statement in the “Confidential to Editor” section, and submit your "Accept" recommendation.

Reviewer #1: All comments have been addressed

Reviewer #2: All comments have been addressed

2. Is the manuscript technically sound, and do the data support the conclusions?

Reviewer #1: Yes

Reviewer #2: Yes

3. Has the statistical analysis been performed appropriately and rigorously? 

Reviewer #1: N/A

Reviewer #2: Yes

4. Have the authors made all data underlying the findings in their manuscript fully available?

Reviewer #1: Yes

Reviewer #2: Yes

5. Is the manuscript presented in an intelligible fashion and written in standard English?

Reviewer #1: Yes

Reviewer #2: Yes

6. Review Comments to the Author

Reviewer #1: The authors addressed all the points raised, and the manuscript can now be accepted. In the response the authors specified the changes made.

Reviewer #2: I want to thank the author of this submission to address my concerns and answering my questions about the technical details of the analysis.

7. PLOS authors have the option to publish the peer review history of their article (what does this mean?). If published, this will include your full peer review and any attached files.

Reviewer #1: No

Reviewer #2: No

---

## [Editor Report · Acceptance letter]

20 Jul 2022

PONE-D-21-36117R1 

The network signature of constellation line figures 

Dear Dr. Bucur:

I'm pleased to inform you that your manuscript has been deemed suitable for publication in PLOS ONE. Congratulations! Your manuscript is now with our production department. 

Kind regards, 

on behalf of

Dr. Diego Raphael Amancio 

Academic Editor

PLOS ONE